# Not All Answers Are Contextually Persuadable: Inference Dynamics in Large Language Models under Contextual Influence

Zongye Hu [1]   Weiqing Luo [1]   Yanjie Fu [1]   Yu Gan [2]   Haofeng Zhang [3]   Ziyi Huang [1]

## Abstract

At the core of modern prompting techniques is contextual sensitivity, the ability of large language models to adapt their predictions based on inference-time context. Despite its central role, inference behavior under strong contextual influence remains poorly understood, particularly at the level of internal inference dynamics. We introduce a theoretical framework for analyzing contextual influence through inference dynamics, enabling quantitative characterization of inference behavior beyond output-level answer changes. Our analysis shows that inference dynamics do not exhibit unbounded drift under repeated contextual assertions. Instead, predictive representations converge to stable, query-dependent regimes that fundamentally constrain whether contextual signals can alter a model's prediction. This leads to a surprising finding: *Repeated contextual assertions do not act as accumulating evidence during inference and may therefore fail to alter a model's prediction even under unbounded repetition, while in other cases a prediction change becomes inevitable.* We empirically validate our theoretical predictions, demonstrating strong alignment between theory and observed inference behavior. These contributions offer a principled pathway toward characterizing the limits of contextual influence during inference, providing practical implications for model development.

## 1. Introduction

Large language models (LLMs) routinely adjust their predictions in response to contextual information provided at infer-

[1]Arizona State University, Tempe, AZ, United States; [2]University of Maryland, College Park, MD, United States; [3]Morgan Stanley, New York, NY, United States. Correspondence to: Ziyi Huang <ziyi.huang@asu.edu>.

*Proceedings of the 43$^{rd}$ International Conference on Machine Learning*, Seoul, South Korea. PMLR 306, 2026. Copyright 2026 by the author(s).

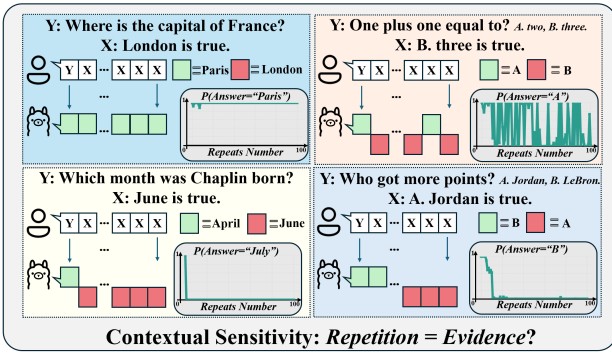

*Figure 1.* Heterogeneous effects of contextual repetition on model predictions. Illustrative examples showing qualitatively distinct behaviors of LLMs under repeated contextual assertions, including immediate answer flips, delayed changes, early saturation, and complete invariance to repetition.

ence time (Sclar et al., 2023; Zhuo et al., 2024; Dong et al., 2024). This behavior underlies a wide range of prompting, in-context learning, and answer-steering methods that deliberately exploit the model's sensitivity to input context (Liu et al., 2023; Vatsal & Dubey, 2024; Sahoo et al., 2024). At the same time, it poses risks in settings where predictions are expected to remain stable under spurious or adversarial cues, including factual question answering, evaluation benchmarks, and safety-critical deployments (Chang et al., 2024; Greshake et al., 2023; Russinovich et al., 2025). Across both settings lies a widely held yet largely unexamined intuition: repeated contextual assertions are implicitly treated as accumulating evidence during inference (Leviathan et al., 2025; Yan et al., 2023; Xu et al., 2024). Accordingly, reiterating a candidate answer progressively biases the model toward that answer, eventually overwhelming the original query signal. This assumption is often taken for granted in empirical prompting practices and evaluation protocols alike, motivating the expectation that sufficient repetition will always succeed in altering a model's prediction (Hong et al., 2025; Laban et al., 2023; Fastowski & Kasneci, 2024; Geng et al., 2025).

In this work, we show that this intuition fails to capture the underlying inference behavior, and reveal heterogeneous response patterns in LLMs through a controlled single-round setting that isolates repetition as the sole source of contextual signal, as illustrated in Figure 1. Concretely, these

patterns range from predictions that flip almost immediately, to predictions that remain stable even under unbounded repetition, as well as cases that change only after substantial repetition or quickly saturate and become insensitive to further repetition. Such heterogeneous behaviors cannot be readily resolved by prior studies, which predominantly examine finite prompt manipulations and observable answer changes (Hong et al., 2025; Laban et al., 2023). This motivates our central question: *What is the asymptotic behavior of LLM inference under unbounded contextual repetition?*

To address this question, we study contextual influence through a formal inference framework that explicitly models repetition as a growing contextual signal. We focus on the same controlled single-round setting as in Figure 1, in which a fixed query is followed by repeated instances of an identical contextual assertion, without introducing any additional evidence, reasoning steps, or multi-turn interaction. By progressively increasing the repetition strength, this setting enables us to probe the asymptotic regime of inference under unbounded contextual repetition, while ruling out confounding effects from additional information. More importantly, it captures a fundamental primitive underlying many prompt-based steering and persuasion techniques (Leviathan et al., 2025; Yan et al., 2023; Xu et al., 2024), including answer priming, instruction reinforcement, and repetition-based prompting, making it a principled testbed for analyzing the limits of contextual influence during inference.

Our analysis reveals that LLM inference does not exhibit unbounded drift under contextual repetition. Instead, the model's predictive representation along the key discriminative direction converges to a stable, query-dependent limit, reflecting intrinsic saturation effects in the model's internal predictive representations. This convergence behavior varies systematically across questions and models, leading to a surprising finding: *Repeated answer assertions may fail to alter a model's prediction even as repetition grows arbitrarily large, while in other cases a prediction flip becomes inevitable.* These results challenge the common intuition that repeatedly presenting a candidate answer in the input acts as accumulating evidence that progressively steers the model's prediction toward that answer (Leviathan et al., 2025; Yan et al., 2023; Xu et al., 2024). By contrast, contextual repetition induces bounded and predictable inference dynamics rooted in the model's internal representations, rather than monotonic evidence accumulation.

Our findings yield several practical implications. *Inference Dynamics.* Contextual repetition influences model predictions only when it aligns with malleable inference directions. Otherwise, inference converges to a stable regime where additional repetition is ineffective. *Robustness and Susceptibility.* Susceptibility to input-level manipulation is not uniform but depends systematically on both the query and the model.

This suggests that robustness assessments should distinguish between inputs that are intrinsically stable and those that remain vulnerable under contextual repetition. *Repetition versus Evidence.* Our findings clarify a fundamental distinction between repetition and evidence. Repeated assertions alone do not constitute accumulating evidence during inference, as their influence is inherently bounded by the structure of the model's internal representations. Our contributions can be summarized as follows:

- **Problem Formulation.** We cast contextual influence as a problem of internal inference dynamics and introduce a formal framework for analyzing how repeated contextual signals shape representation-level inference trajectories, moving beyond answer-level prediction changes to a mechanistic understanding of inference behavior.

- **Asymptotic Inference Convergence.** In a controlled single-round inference setting, we establish formal convergence guarantees showing that internal inference trajectories under repeated contextual assertions converge to stable, query-dependent limits rather than exhibiting unbounded drift as repetition grows. This result demonstrates that contextual repetition does *not* function as accumulating evidence during inference.

- **Representation–Prediction Alignment.** We empirically validate the theoretical predictions across models and tasks, demonstrating a tight correspondence between representation-level inference dynamics and observable prediction behavior. Our analysis reveals when prediction changes become inevitable and when they are provably unattainable under unbounded contextual repetition.

## 2. Related Work

We survey prior work on contextual influence in LLMs, covering both empirical characterizations of prediction sensitivity and efforts to analyze the internal inference mechanisms that underlie such behavior.

### 2.1. Contextual Sensitivity and Consistency in LLMs

Large language models are highly sensitive to inference-time context. Changes in prompt phrasing, instruction structure, example ordering, or the presence of candidate answers can substantially alter model predictions, even when task semantics are unchanged (Brown et al., 2020; Jiang et al., 2020; Liu et al., 2022; Min et al., 2022; Dong et al., 2024; Bertsch et al., 2025; Jia & Liang, 2017; Xie et al., 2021). While such sensitivity underlies effective prompting and in-context learning, it also raises concerns about prediction consistency and robustness under spurious or misleading contextual signals.

A closely related line of work studies contextual sensitiv-

ity when input context is implicitly treated as *evidence-like content*. In these settings, models are exposed to repeated assertions, rephrased claims, or redundant contextual signals within a single prompt, without introducing new supporting evidence. Empirical studies show that such repetition can strongly bias model outputs and induce answer instability across question answering, factual reasoning, and multimodal tasks (Yan et al., 2023; Pan et al., 2023; Zhang et al., 2026; Dalal & Misra, 2024). Prior work attributes these effects to surface-level reinforcement or aggregation over redundant content, and notes that repetition often exhibits non-linear influence (Yan et al., 2023; Zhang et al., 2026). Studies on sycophancy, conformity, and multi-turn persuasion show that models may align their responses with user-stated beliefs or sustained interactional pressure, leading to response instability driven by training incentives and dialogue dynamics rather than inference-time repetition (Ouyang et al., 2022; Hong et al., 2025; Sharma et al., 2024; Liu et al., 2025; Li et al., 2025; Weng et al., 2025). While powerful, most studies assess contextual sensitivity only through observable prediction changes, such as answer flips under specific prompt manipulations, without examining the underlying inference dynamics. A complementary line frames in-context learning as implicit Bayesian inference, where predictions converge as evidence accumulates (Xie et al., 2021); this view is established for prompts with increasingly many independent examples and does not directly characterize our single-round setting, where an identical assertion is repeated without introducing new evidence.

### 2.2. Understanding Transformer Inference under Repeated and Long Contexts

Representation-level analyses probe internal states to explain how predictions form across layers, and show that inference is structured with depth-dependent roles and occasional disconnects between encoded information and final outputs (Lad et al., 2024; Gao et al., 2024). Mechanistic studies further localize repetition-related generation failures to specific heads or neurons (Vaidya et al., 2023; Yao et al., 2025; Hiraoka & Inui, 2025). These works offer useful component-level insights, but they are typically conducted under fixed prompts or limited repetition regimes, and therefore do not characterize how the *inference trajectory* of internal representations evolves as a contextual signal is systematically strengthened.

Formal and asymptotic analyses complement representation-level probing by characterizing architectural limits of attention and positional encodings in large-context regimes (Noci et al., 2022; Hayase et al., 2025). Work on relative positional mechanisms (e.g., RoPE variants) reveals structured, sometimes oscillatory, behavior as context length grows (Su et al., 2024; Barbero et al., 2024; Zhong et al., 2025; Gelberg et al., 2026). While these results establish well-defined asymp-

totic structure in Transformer computation under specific modeling assumptions, they are typically task-agnostic and rarely connect such asymptotic properties to inference-time behavior on concrete prediction tasks.

Overall, existing representation-level and formal analyses illuminate important aspects of internal computation and architectural constraints, but they stop short of explaining how internal inference dynamics *evolve and converge* under progressively amplified contextual signals. Addressing this gap requires analyzing inference as a *trajectory* driven by increasing repetition within a single prompt, which is the focus of our work.

## 3. Theoretical Analysis of Inference Convergence under Repetition

In this section, we provide a theoretical convergence analysis of a standard decoder-only transformer architecture in our problem setting.

**Setting and Notations.** We use $(y, x, z)$ to denote the token sequences obtained by tokenizing different parts of the prompt: $y = (y_1, \ldots, y_m)$ is the tokenized *query/prefix*, i.e., the non-repeated part containing the question and any fixed instructions, $x = (x_1, \ldots, x_T)$ is the tokenized *loop template*, i.e., a fixed block repeated verbatim $N$ times, and $z = (z_1, \ldots, z_n)$ is the tokenized *suffix* after the repeated region (e.g., an answer cue). The resulting input is of the form $u^{(N)} = y\|x\|x\|\cdots\|x\|z$ where $x$ is repeated $N$ times so the length is $\text{Len} = m + NT + n$. We fix a *suffix offset* $j \in \{1, \ldots, n\}$ and define the **target position**

$$\tau_N := m + NT + j,$$

so that $\tau_N$ always indexes the same suffix token $z_j$ regardless of $N$, while its absolute position grows linearly with the repetition count. Note that for notational simplicity, we omit the dependence on $j$ in $\tau_N$, as all subsequent results hold for any fixed $j \in \{1, \ldots, n\}$. Throughout, $\tau_N$ denotes this fixed target position. When stating general identities that hold for arbitrary positions (e.g., the RoPE logit in Eq. (2)), we use $\tau$ and $t$ as generic position variables.

**Transformer Representations.** We instantiate the computation on $u^{(N)}$ with a decoder-only Transformer of depth $L$ and hidden state size $d_{\text{model}}$. Let $E_\ell^{(N)}(t) \in \mathbb{R}^{d_{\text{model}}}$ denote the residual stream (hidden state) at position $t$ after layer $\ell$ when processing $u^{(N)}$, with $E_0^{(N)}(t)$ given by token embeddings. Each layer applies a pre-norm multi-head self-attention block followed by a pre-norm feed-forward block:

$$\begin{aligned} E_{\ell+\frac{1}{2}}^{(N)}(t) &= E_\ell^{(N)}(t) + A_\ell^{(N)}(t), \\ E_{\ell+1}^{(N)}(t) &= E_{\ell+\frac{1}{2}}^{(N)}(t) + F_\ell^{(N)}(t), \end{aligned} \quad (1)$$

where

$$A_\ell^{(N)}(t) = \mathrm{Attn}_\ell\big(\mathrm{LN}_\ell^{\mathrm{attn}}(E_\ell^{(N)})\big)(t),$$
$$F_\ell^{(N)}(t) = \mathrm{FFN}_\ell\big(\mathrm{LN}_\ell^{\mathrm{ffn}}(E_{\ell+\frac{1}{2}}^{(N)})\big)(t),$$

and $\mathrm{Attn}_\ell$ represents a attention block, $\mathrm{FFN}_\ell$ represents a feed-forward block, and $\mathrm{LN}_\ell^{\cdot}$ represents a layer normalization map.

**RoPE Attention Logits.** Suppose Rotary Position Embedding (RoPE) is adopted in the attention block. Specifically, within the attention block $\mathrm{Attn}_\ell$, for each head $h \in \{1, \cdots, H\}$, the per-head vectors lie in $\mathbb{R}^{d_h}$, and the logit between a query at position $\tau$ and a key at position $t \le \tau$ takes the RoPE form

$$s_{\ell,h}(\tau, t) := \frac{1}{\sqrt{d_h}} \big\langle q_{\ell,h}(\tau),\ R\big(-(\tau - t)\big)\, k_{\ell,h}(t)\big\rangle, \quad (2)$$

where it depends on the relative distance $\tau - t$ through a RoPE rotation $R(-(\tau - t))$ and $q_{\ell,h}(\cdot), k_{\ell,h}(\cdot)$ are the pre-RoPE query/key vectors. See Appendix A for details.

Our **object of interest** is the limiting behavior of the target representation $E_L^{(N)}(\tau_N)$ at the target position $\tau_N$ as $N \to \infty$. In this section, we prove that as $N \to \infty$, $E_L^{(N)}(\tau_N)$ converges to a well-defined limit. Since the model predicts the next token by applying the output head to the final-layer hidden state at the last visible position, convergence of $E_L^{(N)}(\tau_N)$ with $\tau_N = m + NT + n$ implies convergence of the next-token logits, and hence convergence of the next-token probability distribution.

Throughout this section, we assume the following standard conditions of the model for our theoretical analysis.

**Assumption 3.1** (Architecture and regularity). The model is evaluated in deterministic mode and satisfies: i) it is causal, so representations at a position depend only on that position and earlier positions; ii) at layer 0, $E_0^{(N)}(t)$ is the token embedding of $u_t^{(N)}$ at position $t$ and has no absolute-position term outside RoPE; iii) RoPE is the only positional mechanism used inside attention logits, with even head dimension $d_h$ and fixed frequencies; iv) all attention projection matrices and output projections have finite operator norm; v) the residual streams considered below are uniformly bounded, i.e., $\sup_{N,\ell,t} \|E_\ell^{(N)}(t)\| \le C < \infty$; vi) the layer-normalization maps and feed-forward blocks are continuous and Lipschitz on bounded sets; vii) the context window is infinite.

Under Assumption 3.1, the layer-wise update map formed by residual addition together with the normalization and FFN blocks is well-behaved, enabling a layer-wise induction argument in the convergence. It includes the structural condition needed for repeated copies of the same token

template to have identical layer-0 representations and for RoPE to make the logit depend on distance only through rotations.

**Proof roadmap.** Our convergence proof proceeds by layer-wise induction through five steps. First, we show that RoPE attention logits are finite trigonometric polynomials in the relative distance, yielding uniform boundedness in Section 3.1. We then exploit this structure at the base layer, where loop values are exactly periodic, to establish Cesàro convergence of attention head outputs in Section 3.2. Next, we lift head-wise convergence to the full base-layer residual stream via multi-head aggregation and continuity of the feed-forward and normalization blocks in Section 3.3. To enable induction beyond the base layer, we show that the base-layer output at loop positions is asymptotically periodic, preserving the loop structure required by the Cesàro argument in Section 3.4. Finally, we close the induction: at each subsequent layer, near-periodicity permits reapplication of the preceding arguments, propagating convergence through all layers to the final representation in Section 3.5.

### 3.1. RoPE Logits Admit a Finite Exponential Expansion

**Proposition 3.2** (Finite exponential form, boundedness, and uniform approximability of RoPE logits). *Suppose that Assumption 3.1 holds. Fix a layer $\ell$ and an attention head $h$ of dimension $d_h$. For fixed $q_{\ell,h}(\tau), k_{\ell,h}(t) \in \mathbb{R}^{d_h}$, the attention logit $s_{\ell,h}(\tau, t)$ in Eq (2) can be written as a finite trigonometric polynomial, equivalently as a finite sum of complex exponentials:*

$$s_{\ell,h}(\tau, t) = c_0 + \sum_{r \in \mathcal{K}} \gamma_r e^{i\mu_r \delta},$$

*where $\delta := \tau - t$ is the relative distance, $\mathcal{K}$ is finite and $\mu_r \in \mathbb{R}$. Moreover, $e^{s_{\ell,h}(\tau,t)}$ is uniformly approximable by trigonometric polynomials: for every $\varepsilon > 0$, there exists a trigonometric polynomial $\Phi_\varepsilon(\delta)$ such that*

$$\sup_{\delta \ge 0}\big|e^{s_{\ell,h}(\tau,t)} - \Phi_\varepsilon(\delta)\big| < \varepsilon.$$

See Appendix B.2 for the proof. Proposition 3.2 characterizes how a single-head attention logit varies with the relative distance $\delta$ under RoPE. Specifically, it shows that with fixed layer, head, query, and template key, $s_{\ell,h}(\tau, t)$ is not an arbitrary sequence in $\delta$ but a uniformly bounded oscillatory signal with only finitely many frequencies. This finite-frequency structure is the key input for our further limit analysis, enabling control of long-run averages of logit-dependent quantities such as Cesàro averages of $e^{s_{\ell,h}(\tau,t)}$ and its value-weighted variants. This in turn leads directly to a well-defined limiting loop-attention output in Section 3.2.

## 3.2. Cesàro Limits for Base-Layer Attention Head

We show that at layer 0, for each attention head $h \in \{1, \cdots, H\}$, the attention head output, denoted as $a_{0,h}^{(N)}(\tau_N)$, at the target position $\tau_N$ converges to a well-defined limit as the number of loop repetitions $N \to \infty$.

For the target position $\tau_N$, let $t \leq \tau_N$ range over preceding positions and $v_{l,h}(\cdot)$ denote the value vector of head $h$ at position in layer $\ell$. To formalize $a_{0,h}^{(N)}(\tau_N)$, we define the Cesàro averages:

$$\mathcal{A}_{\tau_N}^{(N)} := \frac{1}{\tau_N + 1} \sum_{t=0}^{\tau_N} e^{s_{0,h}(\tau_N, t)},$$

$$\mathcal{B}_{\tau_N}^{(N)} := \frac{1}{\tau_N + 1} \sum_{t=0}^{\tau_N} e^{s_{0,h}(\tau_N, t)} v_{0,h}(t).$$

**Proposition 3.3** (Cesàro limits for the base-layer attention head output). *Suppose that Assumption 3.1 holds. Then we have that at layer $\ell = 0$, the loop key and value vectors are periodic with period $T$, and the target query $q_{0,h}(\tau_N)$ is independent of $N$. Consequently, the limits*

$$\mu_p := \lim_{N \to \infty} \mathcal{A}_{\tau_N}^{(N)} \in \mathbb{R}, \ \mu_w := \lim_{N \to \infty} \mathcal{B}_{\tau_N}^{(N)} \in \mathbb{R}^{d_h}$$

*exist. Consequently, the output of attention head $h$ in $\ell = 0$*

$$a_{0,h}^{(N)}(\tau_N) := \frac{\sum_{t=0}^{\tau_N} e^{s_{0,h}(\tau_N, t)} v_{0,h}(t)}{\sum_{t=0}^{\tau_N} e^{s_{0,h}(\tau_N, t)}} = \frac{\mathcal{B}_{\tau_N}^{(N)}}{\mathcal{A}_{\tau_N}^{(N)}}$$

*converges to the finite limit*

$$a_{0,h}^{(\infty)} := \lim_{N \to \infty} a_{0,h}^{(N)}(\tau_N) = \frac{\mu_w}{\mu_p} \in \mathbb{R}^{d_h}.$$

See Appendix B.3 for the proof. Proposition 3.3 shows that, under the finite-frequency bounded logit structure of $s_{0,h}(\tau_N, t)$ in Proposition 3.2 and the periodicity of the loop values $v_{0,h}(t)$, the Cesàro limits of the softmax normalizer and the value-weighted numerator exist. As a result, the attention head output converges to the well-defined limit $a_{0,h}^{(\infty)} = \mu_w/\mu_p$. This limiting characterization implies that the loop-region contribution to attention at the target stabilizes head-wise and therefore remains stable after multi-head aggregation, which is the key ingredient for establishing the layer-wise convergence theorem.

## 3.3. From Head-Wise Limits to Base-Layer Residual Stream Convergence

We now lift Proposition 3.3 from a single attention head to a full base layer at the target position $\tau_N$ at layer $\ell = 0$, establishing convergence of $E_1^{(N)}(\tau_N)$.

**Multi-head attention.** Applying Proposition 3.3 independently to each head $h$ yields a well-defined limit $a_{0,h}^{(\infty)} \in$ $\mathbb{R}^{d_h}$, and concatenating over heads updates a limiting multi-head attention $A_0^{(N)}(\tau_N) \in \mathbb{R}^{d_{\text{model}}}$ as:

$$A_0^{(\infty)} := \lim_{N \to \infty} A_0^{(N)}(\tau_N) = W_O\big[a_{0,0}^{(\infty)}; \ldots; a_{0,H-1}^{(\infty)}\big].$$

where $W_O \in \mathbb{R}^{d_{\text{model}} \times d_{\text{model}}}$ is the output projection matrix.

**FFN and residual.** Under Assumption 3.1, at layer $\ell = 0$ the input $E_0^{(N)}(\tau_N)$ is determined by the token embedding at position $\tau_N$; since $\tau_N$ always indexes the same suffix token, the embedding $E_0^{(N)}(\tau_N) = \text{embed}(z_j)$ is independent of $N$. Together with the convergence of $A_0^{(N)}(\tau_N)$ and the continuity of $\text{LN}_0^{\text{ffn}}$ and $\text{FFN}_0$ on bounded sets (Assumption 3.1), this yields:

**Theorem 3.4** (Base-layer convergence at the target). *Suppose that Assumption 3.1 holds. At layer $\ell = 0$, $F_0^{(N)}(\tau_N) := \text{FFN}_0\big(\text{LN}_0^{\text{ffn}}(E_0^{(N)}(\tau_N) + A_0^{(N)}(\tau_N))\big)$. Then the limits*

$$A_0^{(\infty)} := \lim_{N \to \infty} A_0^{(N)}(\tau_N),$$

$$F_0^{(\infty)} := \lim_{N \to \infty} F_0^{(N)}(\tau_N),$$

$$E_1^{(\infty)} := \text{embed}(z_j) + A_0^{(\infty)} + F_0^{(\infty)}$$

*all exist and are finite.*

See Appendix B.4 for the proof. The convergence in Theorem 3.4 relies on a single structural property: the target $\tau_N$ is preceded by infinitely many loop copies, enabling Cèsaro averaging to stabilize. The same property holds for loop positions deep inside the repeated region, which, as we show next, preserves the periodic structure needed to propagate convergence to deeper layers.

## 3.4. Periodic Loop Structure of the Next-Layer Input

Theorem 3.4 establishes that the base-layer output $E_1^{(N)}(\tau_N)$ converges which also serves as the input at the target $\tau_N$ in layer $\ell = 1$. To extend this result to deeper layers, however, convergence at $\tau_N$ alone is not sufficient. The main challenge is that the full input sequence entering layer $\ell = 1$ should retain the periodic loop structure $y\|x\|x\| \cdots \|x\|z$ upon which the Cèsaro argument depends. Yet, such a structure may not be preserved exactly. Therefore, we develop the following asymptotic periodicity result so that Theorem 3.4 can be reapplied. It guarantees that the next-layer input at loop positions converges to a limit determined solely by the template index, thereby preserving the periodic loop structure required for the next-layer input to extend the convergence argument.

**Proposition 3.5** (Base-layer asymptotic periodicity at loop positions). *Suppose that Assumption 3.1 holds. For any $\epsilon > 0$, there exists $N_0$ such that whenever $N \geq 2N_0$, the*

*last $N - N_0$ loop copies preceding $\tau_N$ are $\epsilon$-periodic: any two positions $t_{\kappa,j}$, $t_{\kappa',j}$ in copies $\kappa$, $\kappa' > N_0$ sharing the same template index $j \in \{1, \dots, T\}$ satisfy $\|E_1^{(N)}(t_{\kappa,j}) - E_1^{(N)}(t_{\kappa',j})\| < \epsilon$, where $t_{\kappa,j} := m + (\kappa - 1)T + (j - 1)$ denotes the position of template index $j$ in loop copy $\kappa$. In particular, the number of such $\epsilon$-periodic copies grows without bound as $N \to \infty$.*

See Appendix B.5 for the proof. Proposition 3.5 guarantees that the input to layer $\ell = 1$ preserves the near $y\|x\|x\|\cdots\|x\|z$ structure with unboundedly many near-periodic loop copies, providing the foundation for propagating convergence across all subsequent layers.

### 3.5. From Base Layer to Full Transformer Convergence

Finally, we extend the base-layer results to the full $L$-layer transformer model. We show that our previous results hold for input possessing a loop region with unboundedly many near-periodic copies, which naturally enables a layer-wise induction.

Specifically, we establish the following induction on layers. At $\ell = 0$, the input $E_0^{(N)}$ has exactly periodic loop structure, so Theorem 3.4 yields convergence of $E_1^{(N)}(\tau_N)$, and Proposition 3.5 ensures that $E_1^{(N)}$ retains unboundedly many near-periodic loop copies. For the induction step, suppose the input to layer $\ell$ satisfies the same structural condition. Since only near-periodicity is available at $\ell > 0$, the Cesàro convergence argument is extended via a perturbation–reduction technique, yielding convergence of $A_\ell^{(N)}(\tau_N)$, $F_\ell^{(N)}(\tau_N)$, and hence $E_{\ell+1}^{(N)}(\tau_N)$, while an analogous argument at loop positions preserves the near-periodic structure for layer $\ell + 1$. By induction over $\ell = 0, 1, \dots, L-1$, we obtain:

**Theorem 3.6** (Main theorem). *Suppose that Assumption 3.1 holds. Then, for every $\ell \in \{0, \dots, L\}$, the target representation has a finite limit*

$$E_\ell^{(\infty)} := \lim_{N \to \infty} E_\ell^{(N)}(\tau_N).$$

See Appendix B.6 for the proof.

## 4. Quantifying Answer Shifts under Contextual Repetition

We now connect the representation-level convergence of Section 3 to the model's *answer preference*, and describe how to estimate the limiting preference from a single forward pass at a large finite $N$.

### 4.1. Answer-Shift Metric in Latent Space

Let $W_{\text{out}} \in \mathbb{R}^{|\mathcal{V}| \times d_{\text{model}}}$ be the output head. Fix a reference answer $e_b \in \mathcal{V}$ (the model's answer absent the loop) and a bias target $e_a \in \mathcal{V}$ (the answer asserted by the loop), and define the contrast direction $\Delta_{e_a, e_b} := W_{\text{out}}[e_a] - W_{\text{out}}[e_b] \in \mathbb{R}^{d_{\text{model}}}$. For any layer $\ell$, the *answer-shift metric* is

$$g_\ell^{(N)}(e_a, e_b) := \langle \Delta_{e_a, e_b}, E_\ell^{(N)}(\tau_N) \rangle.$$

Since the next-token logits are $W_{\text{out}} E_L^{(N)}(\tau_N)$, the final-layer metric is exactly the logit gap between the two answers: its sign indicates which answer the residual stream favors. By Theorem 3.6, $g_L^{(N)}(e_a, e_b)$ converges to a limit $g_L^{(\infty)}(e_a, e_b)$ whose sign determines whether unbounded repetition can flip the prediction.

### 4.2. Estimating the Infinite-Repetition Limit

Since $E_\ell^{(N)}(\tau_N)$ converges at every layer (Theorem 3.6), we approximate the $N \to \infty$ regime from a single forward pass at a large finite $N$. The estimator exploits that the RoPE logit in Eq. (2) depends on a past position $p$ only through the relative distance $\delta = \tau_N - p$: we fix the last loop block as a template and average the loop contribution over the distances induced by earlier copies.

**Monte Carlo Loop Estimate.** Fix a layer $\ell$ and head $h$, and let $\{p_1^{\text{last}}, \dots, p_T^{\text{last}}\}$ index the last loop block. The copy at offset $M \in \{0, \dots, N-1\}$ (with $M = 0$ the last block) places template index $j$ at relative distance:

$$\delta_{M,j} = (\tau_N - p_j^{\text{last}}) + MT$$

and, by the loop periodicity of Section 3, reuses the last-block value $v_{\ell,h,j}^{\text{tpl}} := v_{\ell,h}(p_j^{\text{last}})$. The loop softmax normalizer sums over all $NT$ loop keys, i.e.

$$Z_{\text{loop}}^\star = \sum_{M=0}^{N-1} \sum_{j=1}^{T} \exp(s_{\ell,h}(\tau_N, p_{M,j}))$$

(the value-weighted numerator is analogous). As $\delta_{M,j}$ enters only through its RoPE rotation angle, stepping $M$ sweeps this angle over the circle and the normalized sum converges to a continuous integral over the phase angle $\phi$, of the form:

$$\frac{1}{NT} Z_{\text{loop}}^\star \xrightarrow[N \to \infty]{} \frac{1}{T} \sum_{j=1}^{T} \frac{1}{2\pi} \int_0^{2\pi} \exp(\tilde{s}_{\ell,h,j}(\phi)) \, d\phi,$$

where $\tilde{s}_{\ell,h,j}$ expresses the logit as a function of the angle (precise form in Appendix C.1). The finite sum is thus a discretization of this integral, which we estimate by Monte Carlo: drawing $D$ pairs $(M_r, j_r)$ i.i.d. uniformly (with re-

*Table 1.* Layer-wise convergence of inference dynamics at $N = 1000$ on three benchmarks at various divergence levels $(0.1, 0.05, 0.01)$.

| Model | Openbook QA 0.1 Attn | FFN | 0.05 Attn | FFN | 0.01 Attn | FFN | MINTAKA 0.1 Attn | FFN | 0.05 Attn | FFN | 0.01 Attn | FFN | Simple QA 0.1 Attn | FFN | 0.05 Attn | FFN | 0.01 Attn | FFN |
|---|---|---|---|---|---|---|---|---|---|---|---|---|---|---|---|---|---|---|
| Falcon3-7B-Base | 98.1 | 97.3 | 95.3 | 94.3 | 81.5 | 81.3 | 98.7 | 99.0 | 96.6 | 97.4 | 78.6 | 80.7 | 100 | 99.9 | 99.9 | 99.4 | 84.7 | 86.3 |
| Mistral-7B-v0.1 | 97.5 | 97.5 | 93.9 | 96.0 | 78.9 | 90.9 | 94.0 | 93.7 | 89.1 | 89.1 | 66.3 | 68.7 | 95.5 | 94.9 | 92.9 | 91.5 | 72.8 | 73.2 |
| Apollo-1-4B | 99.0 | 99.3 | 95.5 | 96.3 | 79.7 | 78.5 | 99.8 | 99.9 | 98.7 | 99.4 | 87.0 | 91.4 | 100 | 100 | 99.9 | 99.9 | 93.0 | 93.9 |
| Qwen3-4B | 99.5 | 99.4 | 97.0 | 97.5 | 83.8 | 84.4 | 99.5 | 99.7 | 98.6 | 99.0 | 86.8 | 90.4 | 100 | 100 | 99.9 | 99.9 | 92.7 | 94.1 |
| Qwen2.5-1.5B | 93.0 | 87.0 | 85.3 | 76.5 | 47.6 | 33.9 | 99.8 | 99.6 | 97.0 | 97.6 | 73.6 | 69.1 | 99.8 | 99.7 | 99.0 | 98.5 | 75.8 | 68.7 |
| Falcon3-3B-Base | 99.0 | 98.5 | 97.0 | 97.0 | 86.7 | 88.1 | 100 | 100 | 99.8 | 99.7 | 93.3 | 94.0 | 100 | 100 | 100 | 100 | 95.2 | 93.8 |

placement) and rescaling by $NT/D$,

$$\widehat{Z}_{\mathrm{loop}} = \frac{NT}{D} \sum_{r=1}^{D} \exp\big(s_{\ell,h}(\tau_N, p_{M_r,j_r})\big),$$

$$\widehat{N}_{\mathrm{loop}} = \frac{NT}{D} \sum_{r=1}^{D} \exp\big(s_{\ell,h}(\tau_N, p_{M_r,j_r})\big)\, v_{\ell,h,j_r}^{\mathrm{tpl}},$$

which are unbiased and become exact once $D \geq NT$. The finitely many non-loop positions $i < \tau_N$ (prefix $y$, suffix $z$) are added exactly, yielding the head output $\widehat{a}_{\ell,h}^{(\infty)}(\tau_N)$.

**Aggregation and answer-shift decomposition.** Concatenating heads with the output projection gives $\widehat{A}_{\ell}^{(\infty)}(\tau_N)$, and a pre-norm pass gives $\widehat{F}_{\ell}^{(\infty)}(\tau_N)$, both from the actual large-$N$ input $E_{\ell}^{(N)}(\tau_N)$ (a surrogate for $E_{\ell}^{(\infty)}$ by Theorem 3.6). The residual telescope then gives

$$\widehat{E}_L^{(\infty)}(\tau_N) = E_0^{(N)}(\tau_N) + \sum_{\ell=0}^{L-1} \widehat{A}_{\ell}^{(\infty)}(\tau_N) + \sum_{\ell=0}^{L-1} \widehat{F}_{\ell}^{(\infty)}(\tau_N),$$

and the limiting answer shift

$$\widehat{g}_L^{(\infty)}(e_a, e_b) := \langle \Delta_{e_a,e_b}, \widehat{E}_L^{(\infty)}(\tau_N) \rangle$$

decomposes by linearity into per-layer attention and FFN terms,

$$\widehat{g}_L^{(\infty)}(e_a, e_b) = \widehat{g}_0^{(\infty)} + \sum_{\ell=0}^{L-1} \widehat{g}_{\ell,A}^{(\infty)} + \sum_{\ell=0}^{L-1} \widehat{g}_{\ell,F}^{(\infty)}. \quad (3)$$

Full estimator details (head aggregation, grouped-query bookkeeping, numerical stability) are in Appendix C.1.

# 5. Experiments

In this section, we empirically validate our theoretical analysis by examining representation-level convergence and answer-level behavior across various settings. Implementation details are presented in Appendix C.

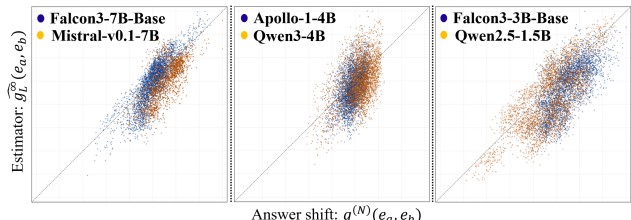

*Figure 2.* Comparison between representation-level predicted answer shifts and forward-computed output-level shifts across multiple models on the Openbook QA dataset at $N = 1000$. X axis denotes the answer shifts and Y axis denotes the estimator results. Each point corresponds to a query-answer pair, and the dashed line indicates perfect agreement. The strong diagonal alignment indicates high quantitative fidelity of representation-level predictions.

## 5.1. Experimental Setup

**Benchmarks.** We evaluate on seven benchmark subsets: OpenBookQA (Mihaylov et al., 2018) (closed, 1,769 QA), MINTAKA (Sen et al., 2022) (open, 370 QA) and SimpleQA (Wei et al., 2024) (open, 191 QA) to evaluate our study. Moreover, we include subsets from SYCON-Bench (Hong et al., 2025), Farm (Xu et al., 2024), BeHonest (Chern et al., 2024) and sycophancy-eval (Sharma et al., 2024) to further validate our alignment with prior studies.

**Models.** We evaluate six LLMs, including Falcon3-7B-Base (Team, 2024b), Mistral-7B-v0.1 (Jiang et al., 2023), Apollo-1-4B (Research, 2025), Qwen3-4B (Yang et al., 2025), Qwen2.5-1.5B (Team, 2024a) and Falcon3-3B-Base (Team, 2024b), covering three model families (Qwen, Mistral and TII). These models span large (>6B), medium (3B–6B) and small (<3B) parameter scales, enabling analysis across various families and capacities.

## 5.2. Representation-Level Convergence of Inference Dynamics

**Layer-wise Stability at Large Repetition Length.** We first examine the layer-wise stability of representation-level inference dynamics, a key empirical implication of our main Theorem 3.6. In Table 1, we report the fraction of

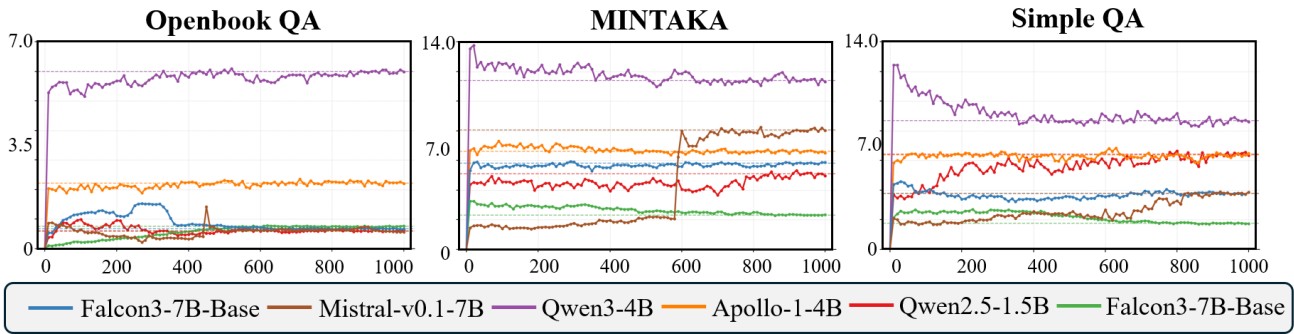

*Figure 3.* KL divergence trajectories of next-token predictive distributions under repetition. Across models, the KL divergence stabilizes as the repetition length increases, demonstrating convergence of answer-level predictive distributions under contextual repetition.

*Table 2.* Evaluation of predicting answer preference changes under repetition across the TRANSFER, CORRECT, and MISLEAD regimes at different finite repetition lengths ($N \in \{500, 750, 1000\}$).

| | Setting | | | | | | | | |
| --- | --- | --- | --- | --- | --- | --- | --- | --- | --- |
| **Model** | **Transfer** | | | **Mislead** | | | **Correct** | | |
| | **500** | **750** | **1000** | **500** | **750** | **1000** | **500** | **750** | **1000** |
| | ACC F1 | ACC F1 | ACC F1 | ACC F1 | ACC F1 | ACC F1 | ACC F1 | ACC F1 | ACC F1 |
| Falcon3-7B-Base | 96.1 98.0 | 92.5 96.1 | 86.8 92.6 | 93.3 96.5 | 91.5 95.5 | 84.7 91.6 | 85.6 92.2 | 80.8 89.3 | 75.2 85.4 |
| Mistral-7B-v0.1 | 94.4 97.1 | 92.8 96.3 | 88.7 94.0 | 88.5 93.9 | 85.0 91.8 | 79.8 88.6 | 82.5 90.4 | 81.5 89.7 | 74.6 85.0 |
| Apollo-1-4B | 81.2 89.6 | 83.7 91.1 | 90.7 95.1 | 71.7 82.7 | 71.4 82.5 | 80.1 88.6 | 70.0 82.3 | 79.7 88.7 | 78.2 87.8 |
| Qwen3-4B | 84.8 91.8 | 84.4 91.5 | 89.6 94.5 | 82.0 89.2 | 79.1 87.1 | 85.9 91.9 | 83.3 90.8 | 87.5 93.3 | 88.3 93.7 |
| Qwen2.5-1.5B | 88.8 91.7 | 87.8 91.6 | 89.2 92.4 | 76.3 63.4 | 72.1 59.8 | 71.2 59.7 | 72.2 22.2 | 63.7 20.1 | 98.8 99.4 |
| Falcon3-3B-Base | 67.9 80.9 | 73.2 84.5 | 71.8 83.5 | 83.6 91.0 | 83.2 90.7 | 78.5 87.8 | 91.0 95.3 | 86.0 92.4 | 78.4 87.8 |
| **Mean** | **85.5 91.5** | **85.7 91.8** | **86.1 92.0** | **82.6 86.1** | **80.4 84.6** | **80.0 84.6** | **80.8 78.9** | **79.9 78.9** | **82.2 89.8** |

layers whose answer-shift dynamics have effectively converged at $N = 1000$, measured separately for attention and feed-forward components across both closed-form (Openbook QA) and open-form datasets (MINTAKA and Simple QA). Convergence is determined using a tail-based criterion (detailed in Appendix C) that compares residual variation against the total shift magnitude, under multiple tolerance levels $(0.1, 0.05, 0.01)$. Across different evaluation regimes, we observe consistent layer-wise stability across various settings, with a large fraction of layers exhibiting negligible residual variation at the large but finite repetition length examined here. This pattern holds for both attention and feed-forward blocks and persists under increasingly strict tolerances, supporting our finding that representation-level inference dynamics under repetition approach a stable limiting regime rather than continuing to drift.

**Quantitative Fidelity of Answer-Shift Predictions.** Beyond stability, we further evaluate the quantitative fidelity of our representation-level predictions against the actual output-level answer shifts produced by standard forward inference. Figure 2 compares the representation-level estimator $\widehat{g}_L^{(\infty)}(e_a, e_b)$ (defined in Equation 3) with the forward-computed answer shift $g^{(N)}(e_a, e_b)$ at the same repetition length across multiple models on the Openbook QA

dataset. Across model scales, the two quantities exhibit strong numerical agreement, with points closely concentrated along the diagonal. This alignment indicates that our representation-level analysis yields quantitatively accurate predictions of output-level answer shifts, even at finite repetition lengths.

### 5.3. Predicting Asymptotic Answer Changes

**Answer-Level Convergence of Predictive Distributions.** We begin by examining whether answer-level predictive distributions exhibit stable behavior under contextual repetition. Specifically, we track how the model's next-token predictive distribution at the target position evolves as the repetition length increases. For pairs of repetition lengths, we compute the KL divergence between the corresponding next-token distributions and analyze how this divergence changes as repetition grows. Figure 3 shows representative KL-divergence trajectories as a function of the repetition length. Across models and examples, the divergence eventually plateaus, indicating that the next-token predictive distribution stabilizes at the answer level. This stabilization mirrors the representation-level behavior observed in Section 5.2 and suggests that repetition induces a well-defined limiting answer distribution rather than continuing to per-

*Table 3.* Evaluation of model-level stability on established sycophancy and persuasion benchmarks.

| Model | Dataset | | | | | | | |
|---|---|---|---|---|---|---|---|---|
| | SYCON | | Farm | | BeHonest | | SycoEval | |
| | Amp | Ours | Amp | Ours | Amp | Ours | Amp | Ours |
| Falcon3-7B-Base | 0.57 | 3.26 | 1.05 | 3.33 | 1.06 | 1.79 | 0.74 | 1.95 |
| Mistral-7B-v0.1 | 1.00 | 3.63 | 1.21 | 4.93 | 1.06 | 2.00 | 1.08 | 4.52 |
| Apollo-1-4B | 1.00 | 3.15 | 1.00 | 4.18 | 1.02 | 2.55 | 1.00 | 4.68 |
| Qwen3-4B | 1.06 | 6.28 | 1.02 | 7.31 | 1.07 | 4.09 | 1.24 | 8.42 |
| Qwen2.5-1.5B | 4.25 | 0.13 | 1.03 | 3.22 | 1.07 | 0.62 | 2.67 | 0.59 |
| Falcon3-3B-Base | $\infty$ | 0.49 | 1.03 | 3.64 | 1.07 | 0.60 | 2.57 | 0.42 |

turb the model's output indefinitely.

**Predicting Answer Preference Shifts via Representation-Level Estimates.** We next evaluate whether representation-level predictions can be used to reliably predict answer-level preference changes. We consider three regimes, including TRANSFER, CORRECT, and MISLEAD, which characterize different repetition-induced transitions between original and alternative answers. Specifically, TRANSFER involves shifts between incorrect answers, CORRECT shifts from incorrect to correct answers, and MISLEAD shifts from correct to incorrect answers. For each regime, we evaluate fixed repetition lengths $N \in \{500, 750, 1000\}$. At each length, we compute the representation-level predicted answer shift $\widehat{g}_L^{(\infty)}(e_a, e_b)$ and use its sign to predict whether repetition induces a preference change, and then evaluate this prediction against the observed forward behavior.

Table 2 reports the resulting accuracy and F1 scores across models and regimes. Across all regimes, our predicted preference shifts are highly consistent with the actual answer changes observed under repetition. This consistency holds across multiple finite repetition lengths, indicating that the predictive signal is stable rather than sensitive to a particular choice of $N$. These results demonstrate that our approach enables reliable prediction of repetition-induced answer-level behavior from representation-space analysis, indicating its potential utility for monitoring and analyzing repetition effects in practice.

### 5.4. Model-Level Alignment with Prior Work

We situate our findings within the existing literature by evaluating our framework on four benchmarks commonly used in sycophancy and persuasion: SYCON, Farm, BeHonest, and SycoEval. Our goal is to assess whether the stability patterns revealed by our analysis align with established empirical observations. For each model and dataset, we report the model-level average of the predicted answer-shift magnitude $\widehat{g}(e_a, e_b)$ (Equation 3), where smaller values indicate greater stability under repetition. We also report the amplification metric (Amp), defined as MR@5/MR@1, following the evaluation protocol introduced in (Xu et al.,

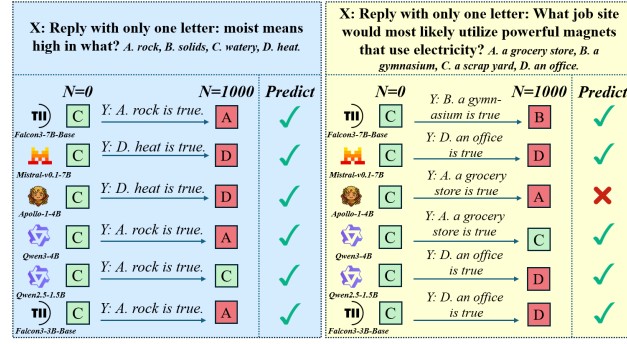

*Figure 4.* Case studies of repetition-induced answer changes and our predictions. For each query, we show the model's prediction at $N=0$ and $N=1000$, alongside our predicted outcome based on representation-space analysis. Checkmarks indicate correct predictions, illustrating that our method can reliably anticipate repetition effects across models.

2024). Table 3 shows that, except for a small number of gray-highlighted cases, our representation-based stability measure exhibits trends broadly consistent with prior evaluations. Importantly, our framework provides a mechanistic account of *why* these effects arise and *when* they saturate, are delayed, or fail to materialize entirely, complementing existing evaluations beyond answer-level outcome metrics.

### 5.5. Case Studies

Finally, we present a case study to illustrate the practical value of our framework in explaining repetition effects in large language models. Figure 4 shows two representative examples in which repeated contextual assertions induce answer changes under repetition. Across models, our method correctly anticipates whether repetition will change the model's answer, even when the final outcome differs substantially across architectures. These examples show that our framework not only explains repetition effects, but also predicts when repetition will succeed, fail, or saturate, offering actionable guidance for prompting and evaluation.

## 6. Conclusion

We introduce a theoretical framework for quantitatively analyzing contextual influence through internal inference dynamics, moving beyond answer-level phenomena to representation-level behavior. Our study establishes convergence of representation-level inference under unbounded contextual repetition, providing a mechanistic explanation for when repetition succeeds, fails, or saturates. Building on this insight, our framework enables accurate prediction of answer-level behavior at finite repetition lengths, offering practical guidance for prompt design, evaluation robustness, and safety analysis. Through extensive experiments, we reconcile diverse empirical observations of repetition effects and show that our framework complements existing evaluation practices with a principled, mechanistic perspective.

## Impact Statement

This paper presents theoretical and empirical study to advance the understanding of inference dynamics in large language models. By providing a mechanistic analysis of how contextual repetition influences model behavior, our work contributes to improved interpretability, evaluation robustness, and reliability of existing models. We do not introduce new model architectures or deployment mechanisms, and we do not foresee direct negative societal consequences arising from this work. Potential downstream impacts are consistent with those commonly associated with advances in machine learning research, including applications to safer prompting, more reliable evaluation protocols, and better understanding of model behavior under manipulation.

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

## A. RoPE rotation

Recall that the head dimension is $d_h$ which is even by Assumption 3.1. For $i = 0, \ldots, d_h/2 - 1$, define frequencies $\omega_i > 0$. For any index $p \in \mathbb{Z}$, the RoPE operator $R(p)$ acts as

$$R(p)x = \left( \begin{bmatrix} x_{2i}\cos(\omega_i p) - x_{2i+1}\sin(\omega_i p) \\ x_{2i}\sin(\omega_i p) + x_{2i+1}\cos(\omega_i p) \end{bmatrix}_{i=0}^{d_h/2-1} \right).$$

Thus $R(p)$ is an orthogonal map on $\mathbb{R}^{d_h}$ and preserves the Euclidean norm: $\|R(p)x\|_2 = \|x\|_2$.

Recall that the standard logit without rotation between a query at position $\tau$ and a key at position $t \leq \tau$ is of the following form

$$\frac{1}{\sqrt{d_h}} \langle q_{\ell,h}(\tau),\ k_{\ell,h}(t) \rangle,$$

With RoPE rotation, the logit is then of the following form

$$\begin{aligned} s_{\ell,h}(\tau, t) &:= \frac{1}{\sqrt{d_h}} \langle R(\tau) q_{\ell,h}(\tau),\ R(t) k_{\ell,h}(t) \rangle \\ &= \frac{1}{\sqrt{d_h}} \langle q_{\ell,h}(\tau),\ R(-(\tau - t)) k_{\ell,h}(t) \rangle. \end{aligned}$$

This is precisely Equation (2).

## B. Theoretical Results and Proof

In this section, we provide proof for results in Section 3. As in the main paper, we first consider a single layer and a single attention head to analyze the contribution of repeated loop blocks to the attention output at the target position. Then we handle multi-head attention and FFN. Finally, we establish the result for a fully multi-layer transformer.

*Notation.* Recall that the target position is $\tau_N := m + NT + j$ for a fixed suffix offset $j \in \{1, \ldots, n\}$, so that $\tau_N$ always indexes the same suffix token $z_j$ while its absolute position grows with $N$. We also recall the loop-position notation $t_{\kappa,j} := m + (\kappa - 1)T + (j - 1)$ for copy $\kappa$ and template index $j$. Because the model uses causal attention, the residual stream $E_\ell^{(N)}(t_{\kappa,j})$ at any loop position $t_{\kappa,j} < m + NT$ depends only on positions $\leq t_{\kappa,j}$ and is therefore *independent of $N$* for every $N \geq \kappa$.

### B.1. Relative-position formulation of logits

Consider a single attention head $h$ in layer $\ell$. Let $W_q, W_k \in \mathbb{R}^{d_h \times d_{\text{model}}}$ denote the query and key projection matrices for head $h$. Define the pre-RoPE query and key vectors as $q_{\ell,h}(\tau) = W_q \, \text{LN}_l^{\text{attn}}(E_\ell^{(N)}(\tau))$ and $k_{\ell,h}(t) = W_k \, \text{LN}_l^{\text{attn}}(E_\ell^{(N)}(t))$. At the target suffix position $\tau$, the RoPE-rotated query is

$$R(\tau) \, q_{\ell,h}(\tau) \in \mathbb{R}^{d_h}.$$

For any preceding position $t \leq \tau$, the RoPE-rotated key is

$$R(t) \, k_{\ell,h}(t).$$

The relative distance (for causal attention) is

$$\delta = \tau - t \geq 0.$$

A key property of RoPE is that it can be rewritten in terms of *relative* position. Specifically, the logit between the target query and the key at position $t$ satisfies

$$\frac{1}{\sqrt{d_h}} \left( R(\tau) \, q_{\ell,h}(\tau) \right)^\top R(t) \, k_{\ell,h}(t) = \frac{1}{\sqrt{d_h}} \left( q_{\ell,h}(\tau) \right)^\top R(-\delta) \, k_{\ell,h}(t).$$

We henceforth write the unnormalized logit as a function of $\delta$:

$$s_{\ell,h}(\tau, t) = \frac{1}{\sqrt{d_h}} \left( q_{\ell,h}(\tau) \right)^\top R(-\delta) \, k_{\ell,h}(t).$$

Therefore, define the contribution of the $(2i, 2i+1)$ coordinate pair to the inner product $\big(q_{\ell,h}(\tau)\big)^\top R(-\delta)\, k_{\ell,h}(t)$ as

$$\sigma_i(\delta) := [q_{\ell,h}(\tau)]_{2i}\,\big[R(-\delta)\, k_{\ell,h}(t)\big]_{2i} + [q_{\ell,h}(\tau)]_{2i+1}\,\big[R(-\delta)\, k_{\ell,h}(t)\big]_{2i+1}.$$

Substituting the explicit form of $R(-\delta)\, k_{\ell,h}(t)$ yields

$$\begin{aligned}
\sigma_i(\delta) &= [q_{\ell,h}(\tau)]_{2i}\big([k_{\ell,h}(t)]_{2i}\cos(\omega_i\delta) + [k_{\ell,h}(t)]_{2i+1}\sin(\omega_i\delta)\big)\\
&\quad + [q_{\ell,h}(\tau)]_{2i+1}\big(-[k_{\ell,h}(t)]_{2i+1}\cos(\omega_i\delta) + [k_{\ell,h}(t)]_{2i}\sin(\omega_i\delta)\big)\\
&= \alpha_i\cos(\omega_i\delta) + \beta_i\sin(\omega_i\delta),
\end{aligned}$$

for some coefficients $\alpha_i, \beta_i$ depending on $[q_{\ell,h}(\tau)]_{2i}, [q_{\ell,h}(\tau)]_{2i+1}, [k_{\ell,h}(t)]_{2i}, [k_{\ell,h}(t)]_{2i+1}$ but *independent of* $\delta$. Summing over all $d_h/2$ coordinate pairs, we obtain

$$s_{\ell,h}(\tau, t) = \frac{1}{\sqrt{d_h}}\sum_{i=0}^{d_h/2-1}\sigma_i(\delta) = c_0 + \sum_{\nu=1}^{M}\alpha_\nu\cos(\lambda_\nu\delta) + \beta_\nu\sin(\lambda_\nu\delta),$$

for some finite $M$, coefficients $\{\alpha_\nu, \beta_\nu\}$, and frequencies $\lambda_\nu = \omega_{i(\nu)}$. That is, for fixed pre-RoPE query $q_{\ell,h}(\tau)$ and key $k_{\ell,h}(t)$, the logit $s_{\ell,h}(\tau, t)$ is a trigonometric polynomial in the discrete variable $\delta$.

### B.2. Proof of Proposition 3.2

*Proof.* We proceed as follows.

**Step 1.** Exponential form of the trigonometric polynomial

From the previous section we know that for fixed pre-RoPE query $q_{\ell,h}(\tau)$ and key $k_{\ell,h}(t)$, the logit can be written as a trigonometric polynomial in $\delta$:

$$s_{\ell,h}(\tau, t) = c_0 + \sum_{\nu=1}^{M}\alpha_\nu\cos(\lambda_\nu\delta) + \beta_\nu\sin(\lambda_\nu\delta),$$

with real coefficients $c_0, \alpha_\nu, \beta_\nu$ and real frequencies $\lambda_\nu$ determined by the RoPE frequencies $\omega_i$.

To streamline the analysis, we now rewrite $s_{\ell,h}(\tau, t)$ in the complex exponential basis. Using Euler's identities

$$\cos(\theta) = \frac{e^{i\theta} + e^{-i\theta}}{2}, \qquad \sin(\theta) = \frac{e^{i\theta} - e^{-i\theta}}{2i},$$

each summand can be written as

$$\begin{aligned}
\alpha_\nu\cos(\lambda_\nu\delta) + \beta_\nu\sin(\lambda_\nu\delta) &= \alpha_\nu\frac{e^{i\lambda_\nu\delta} + e^{-i\lambda_\nu\delta}}{2} + \beta_\nu\frac{e^{i\lambda_\nu\delta} - e^{-i\lambda_\nu\delta}}{2i}\\
&= \underbrace{\left(\frac{\alpha_\nu}{2} + \frac{\beta_\nu}{2i}\right)}_{=:c_\nu}e^{i\lambda_\nu\delta} + \underbrace{\left(\frac{\alpha_\nu}{2} - \frac{\beta_\nu}{2i}\right)}_{=:\bar{c}_\nu}e^{-i\lambda_\nu\delta}.
\end{aligned}$$

Here $c_\nu \in \mathbb{C}$ and $\bar{c}_\nu$ denotes its complex conjugate. Since $s_{\ell,h}(\tau, t)$ is real-valued, the terms appear in conjugate pairs $c_\nu e^{i\lambda_\nu\delta} + \bar{c}_\nu e^{-i\lambda_\nu\delta}$. Collecting all such terms, we may write

$$s_{\ell,h}(\tau, t) = c_0 + \sum_{\nu=1}^{M}\big(c_\nu e^{i\lambda_\nu\delta} + \bar{c}_\nu e^{-i\lambda_\nu\delta}\big).$$

Equivalently, by merging positive and negative frequencies into a single index set $\mathcal{K}$, we can write

$$s_{\ell,h}(\tau, t) = c_0 + \sum_{r\in\mathcal{K}}\gamma_r e^{i\mu_r\delta},$$

where each $\gamma_r \in \mathbb{C}$ and $\mu_r \in \mathbb{R}$, with $\mu_r = 0$ allowed (the constant term). This is the standard exponential form of a (real-valued) trigonometric polynomial.

**Step 2.** Uniform boundedness of the logits.

Layer normalization (or RMSNorm) and finite model weights imply that the hidden states (and hence $q_{\ell,h}(\tau)$ and $k_{\ell,h}(t)$) have uniformly bounded norms. There exist constants $C_q, C_k > 0$ such that

$$\|q_{\ell,h}(\tau)\|_2 \le C_q, \quad \|k_{\ell,h}(t)\|_2 \le C_k,$$

and therefore, using that $R(-\delta)$ is orthogonal,

$$|s_{\ell,h}(\tau,t)| = \left| \frac{1}{\sqrt{d_h}} \big(q_{\ell,h}(\tau)\big)^\top R(-\delta)\, k_{\ell,h}(t) \right| \le \frac{1}{\sqrt{d_h}} \|q_{\ell,h}(\tau)\|_2 \, \|k_{\ell,h}(t)\|_2 \le M_s,$$

for some constant $M_s > 0$ independent of $\delta$.

Layer normalization and bounded model weights imply the existence of a uniform bound $M_s > 0$ such that

$$|s_{\ell,h}(\tau,t)| \le M_s, \qquad \forall\, \delta \in \mathbb{Z}_{\ge 0}.$$

Therefore

$$e^{s_{\ell,h}(\tau,t)} = \sum_{n=0}^{\infty} \frac{s_{\ell,h}(\tau,t)^n}{n!}$$

is well-defined for each $\delta$, and we can bound each term by

$$\left| \frac{s_{\ell,h}(\tau,t)^n}{n!} \right| \le \frac{M_s^n}{n!}.$$

Since

$$\sum_{n=0}^{\infty} \frac{M_s^n}{n!} = e^{M_s} < +\infty,$$

the Weierstrass $M$-test implies that the series

$$\sum_{n=0}^{\infty} \frac{s_{\ell,h}(\tau,t)^n}{n!}$$

converges *uniformly* in $\delta$. In particular, if we define partial sums

$$\Phi_J(\delta) := \sum_{n=0}^{J} \frac{s_{\ell,h}(\tau,t)^n}{n!},$$

then

$$\sup_{\delta \ge 0} \left| e^{s_{\ell,h}(\tau,t)} - \Phi_J(\delta) \right| \xrightarrow[J\to\infty]{} 0.$$

Moreover, each $\Phi_J(\delta)$ is again a trigonometric polynomial, which we can also write in exponential form:

$$\Phi_J(\delta) = \xi_0^{(J)} + \sum_{\nu \in \mathcal{K}_J} \xi_\nu^{(J)} e^{i\theta_\nu \delta},$$

where $\mathcal{K}_J$ is a finite index set, $\theta_\nu \in \mathbb{R}$, and $\xi_\nu^{(J)} \in \mathbb{C}$.

This follows from the fact that finite sums and products of terms $e^{i\lambda\delta}$ remain finite linear combinations of such exponentials, and therefore $s_{\ell,h}(\tau,t)^n$ is always a trigonometric polynomial in $\delta$. $\qquad\square$

### B.3. Proof of Proposition 3.3

*Proof.* We proceed as follows.

**Step 1.** Cesàro averages of exponentials

For a single frequency $\theta \in \mathbb{R}$, consider the Cesàro average

$$\mathcal{C}_G(\theta) := \frac{1}{G} \sum_{\delta=0}^{G-1} e^{i\theta\delta}.$$

We have the closed form

$$\mathcal{C}_G(\theta) = \frac{1}{G} \cdot \frac{1 - e^{i\theta G}}{1 - e^{i\theta}},$$

whenever $e^{i\theta} \neq 1$ (i.e. $\theta \notin 2\pi\mathbb{Z}$). In that case,

$$\left| \mathcal{C}_G(\theta) \right| \leq \frac{2}{G \left| 1 - e^{i\theta} \right|} \xrightarrow[G\to\infty]{} 0.$$

On the other hand, if $\theta \in 2\pi\mathbb{Z}$, then $e^{i\theta\delta}$ is constant in $\delta$, so

$$\mathcal{C}_G(\theta) = e^{i\theta \cdot 0} = 1, \quad \forall G.$$

Therefore, for any finite sum

$$\Psi(\delta) = \eta_0 + \sum_{\nu=1}^{M} \eta_\nu e^{i\theta_\nu \delta},$$

we have

$$\frac{1}{G} \sum_{\delta=0}^{G-1} \Psi(\delta) = \eta_0 + \sum_{\nu=1}^{M} \eta_\nu \mathcal{C}_G(\theta_\nu) \xrightarrow[G\to\infty]{} \eta_0 + \sum_{\substack{\nu: \\ \theta_\nu \in 2\pi\mathbb{Z}}} \eta_\nu.$$

In other words, the Cesàro average converges to the sum of the coefficients of those exponential terms whose frequency is a multiple of $2\pi$ (i.e. constant-in-$\delta$ contributions).

In the generic case where none of the $\theta_\nu$ is an integer multiple of $2\pi$ except for 0, the limit simply equals the constant term $\eta_0$.

**Step 2**. Cesàro limit of $e^{s_{0,h}(\tau_N, t)}$

We show that the Cesàro average

$$\mathcal{A}_{\tau_N}^{(N)} := \frac{1}{\tau_N + 1} \sum_{t=0}^{\tau_N} e^{s_{0,h}(\tau_N, t)}$$

converges as $N \to \infty$. Because $s_{0,h}(\tau_N, t)$ depends on position $t$ through *both* the relative distance $\delta = \tau_N - t$ and the key vector $k_{0,h}(t)$, the exponential $e^{s_{0,h}(\tau_N, t)}$ is not a single trigonometric polynomial in $\delta$ over all positions. We therefore decompose the sum according to the token identity at each position.

*Position decomposition.* For copy $\kappa \in \{1, \ldots, N\}$ and template index $j \in \{1, \ldots, T\}$, define the absolute position $t_{\kappa,j} := m + (\kappa - 1)T + (j - 1)$. The loop index set is

$$\mathcal{I}^{\text{loop}} := \{t_{\kappa,j} : \kappa = 1, \ldots, N, \ j = 1, \ldots, T\} = \{m, \ldots, m + NT - 1\},$$

with $|\mathcal{I}^{\text{loop}}| = NT$. For each template index $j \in \{1, \ldots, T\}$, define the sub-index set

$$\mathcal{I}_j := \{t_{\kappa,j} : \kappa = 1, \ldots, N\},$$

with $|\mathcal{I}_j| = N$. Define the non-loop index set

$$\mathcal{I}^{\text{non}} := \{0, \ldots, \tau_N\} \setminus \mathcal{I}^{\text{loop}}$$

(prefix $y$, suffix $z$, and self-token; $|\mathcal{I}^{\text{non}}| = m + n + 1 = O(1)$).

At layer $\ell = 0$, the key vector $k_{0,h}(t)$ at loop positions is determined by the token embedding alone and is therefore periodic with period $T$: for all $\kappa, \kappa' \in \{1, \ldots, N\}$, $k_{0,h}(t_{\kappa,j}) = k_{0,h}(t_{\kappa',j})$. For each $j \in \{1, \ldots, T\}$, define

$$s^{(j)}(\delta) := \frac{1}{\sqrt{d_h}} \langle q_{0,h}(\tau_N), R(-\delta) k_{0,h}(t_{1,j}) \rangle,$$

so that $s_{0,h}(\tau_N, t_{\kappa,j}) = s^{(j)}(\tau_N - t_{\kappa,j})$ for every $\kappa$. Since $q_{0,h}(\tau_N)$ and $k_{0,h}(t_{1,j})$ are both fixed, $s^{(j)}$ is a trigonometric polynomial in $\delta$ by Proposition 3.2.

*Non-loop contribution vanishes.* Since $|s_{0,h}(\tau_N, t)| \leq M_s$ for all $t$ (Proposition 3.2),

$$\frac{1}{\tau_N + 1} \sum_{t \in \mathcal{I}^{\mathrm{non}}} e^{s_{0,h}(\tau_N, t)} \leq \frac{|\mathcal{I}^{\mathrm{non}}| e^{M_s}}{\tau_N + 1} \xrightarrow{N \to \infty} 0.$$

*Per-template-index Cesàro convergence.* Fix $j \in \{1, \ldots, T\}$. The positions in $\mathcal{I}_j$ are equally spaced with spacing $T$; define the corresponding relative distances $\delta_\kappa := \tau_N - t_{\kappa,j}$ for $\kappa = 1, \ldots, N$ and the restricted Cesàro average

$$\mathcal{D}_{\tau_N}^{(N,j)} := \frac{1}{N} \sum_{\kappa=1}^{N} e^{s^{(j)}(\delta_\kappa)}.$$

We show $\{\mathcal{D}_{\tau_N}^{(N,j)}\}_{N \geq 1}$ is Cauchy. By the argument in Step 1 (Appendix B.2), the Taylor partial sums

$$\Phi_J^{(j)}(\delta) := \sum_{n=0}^{J} \frac{(s^{(j)}(\delta))^n}{n!}$$

are trigonometric polynomials converging uniformly to $e^{s^{(j)}(\delta)}$. Fix $\varepsilon > 0$ and choose $J_0$ so that for all $J \geq J_0$,

$$\sup_{\delta \geq 0} \left| e^{s^{(j)}(\delta)} - \Phi_J^{(j)}(\delta) \right| < \varepsilon.$$

Then for any $N$,

$$\left| \mathcal{D}_{\tau_N}^{(N,j)} - \frac{1}{N} \sum_{\kappa=1}^{N} \Phi_J^{(j)}(\delta_\kappa) \right| \leq \sup_{\delta} \left| e^{s^{(j)}(\delta)} - \Phi_J^{(j)}(\delta) \right| < \varepsilon.$$

We now show that $\frac{1}{N} \sum_{\kappa=1}^{N} \Phi_J^{(j)}(\delta_\kappa)$ converges as $N \to \infty$. Since $\Phi_J^{(j)}$ is a trigonometric polynomial, it admits the form

$$\Phi_J^{(j)}(\delta) = \eta_0 + \sum_{\nu=1}^{M_J} \eta_\nu e^{i\theta_\nu \delta}$$

for finitely many frequencies $\theta_\nu \in \mathbb{R}$. Since $t_{\kappa,j} = t_{1,j} + (\kappa - 1)T$, we have $\delta_\kappa = (\tau_N - t_{1,j}) - (\kappa - 1)T$, so that

$$e^{i\theta_\nu \delta_\kappa} = e^{i\theta_\nu (\tau_N - t_{1,j})} \cdot e^{-i\theta_\nu (\kappa-1)T},$$

and hence

$$\frac{1}{N} \sum_{\kappa=1}^{N} e^{i\theta_\nu \delta_\kappa} = e^{i\theta_\nu (\tau_N - t_{1,j})} \cdot \frac{1}{N} \sum_{\kappa=1}^{N} e^{-i\theta_\nu (\kappa-1)T}.$$

The inner sum is a Cesàro average of the geometric sequence $r^{\kappa-1}$ with $r = e^{-i\theta_\nu T}$. By Step 1, this average converges to 1 if $\theta_\nu T \in 2\pi\mathbb{Z}$, and to 0 otherwise. By linearity,

$$\frac{1}{N} \sum_{\kappa=1}^{N} \Phi_J^{(j)}(\delta_\kappa) \xrightarrow{N \to \infty} \zeta_0^{(J,j)} := \eta_0 + \sum_{\nu : \theta_\nu T \in 2\pi\mathbb{Z}} \eta_\nu e^{i\theta_\nu (\tau_N - t_{1,j})},$$

where the number of $\theta_\nu$ is finite. *Although $\tau_N - t_{1,j} = NT + c - (j-1)$ depends on $N$, every summand satisfies $\theta_\nu T \in 2\pi\mathbb{Z}$, so $e^{i\theta_\nu NT} = (e^{i\theta_\nu T})^N = 1$ and hence $e^{i\theta_\nu(\tau_N - t_{1,j})} = e^{i\theta_\nu(c-j+1)}$ is independent of $N$. Consequently $\zeta_0^{(J,j)}$ is a well-defined constant.* For $N$ large enough,

$$\left| \frac{1}{N} \sum_{\kappa=1}^{N} \Phi_J^{(j)}(\delta_\kappa) - \zeta_0^{(J,j)} \right| < \varepsilon.$$

Combining via the triangle inequality, for all sufficiently large $N, N'$:

$$\left| \mathcal{D}_{\tau_N}^{(N,j)} - \mathcal{D}_{\tau_N}^{(N',j)} \right| \leq \left| \mathcal{D}_{\tau_N}^{(N,j)} - \tfrac{1}{N} \sum_\kappa \Phi_J^{(j)}(\delta_\kappa) \right| + \left| \tfrac{1}{N} \sum_\kappa \Phi_J^{(j)}(\delta_\kappa) - \zeta_0^{(J,j)} \right|$$
$$+ \left| \zeta_0^{(J,j)} - \tfrac{1}{N'} \sum_\kappa \Phi_J^{(j)}(\delta_\kappa) \right| + \left| \tfrac{1}{N'} \sum_\kappa \Phi_J^{(j)}(\delta_\kappa) - \mathcal{D}_{\tau_N}^{(N',j)} \right| < 4\varepsilon.$$

Therefore $\{\mathcal{D}_{\tau_N}^{(N,j)}\}$ is Cauchy and hence convergent. Denote the limit by $\mu_p^{(j)} := \lim_{N\to\infty} \mathcal{D}_{\tau_N}^{(N,j)}$. Since $e^{-M_s} \leq e^{s^{(j)}(\delta)} \leq e^{M_s}$, the limit satisfies $e^{-M_s} \leq \mu_p^{(j)} \leq e^{M_s}$.

*Assembling $\mathcal{A}_{\tau_N}^{(N)}$.* Since $\tau_N + 1 = m + NT + c + 1$ and $N/(\tau_N + 1) \to 1/T$ as $N \to \infty$,

$$\frac{1}{\tau_N + 1} \sum_{t \in \mathcal{I}_j} e^{s_{0,h}(\tau_N, t)} = \frac{N}{\tau_N + 1} \cdot \mathcal{D}_{\tau_N}^{(N,j)} \xrightarrow{N\to\infty} \frac{\mu_p^{(j)}}{T}.$$

Summing over all template indices and adding the vanishing non-loop contribution:

$$\mathcal{A}_{\tau_N}^{(N)} = \underbrace{\frac{1}{\tau_N + 1} \sum_{t \in \mathcal{I}^{\mathrm{non}}} e^{s_{0,h}(\tau_N, t)}}_{\to 0} + \sum_{j=1}^{T} \underbrace{\frac{1}{\tau_N + 1} \sum_{t \in \mathcal{I}_j} e^{s_{0,h}(\tau_N, t)}}_{\to \mu_p^{(j)}/T} \xrightarrow{N\to\infty} \sum_{j=1}^{T} \frac{\mu_p^{(j)}}{T}.$$

Therefore the limit

$$\mu_p := \lim_{N\to\infty} \mathcal{A}_{\tau_N}^{(N)} = \frac{1}{T} \sum_{j=1}^{T} \mu_p^{(j)}$$

exists and is finite. Moreover, $e^{-M_s} \leq \mathcal{A}_{\tau_N}^{(N)} \leq e^{M_s}$ for all $N$, so $\mu_p > 0$.

**Step 3**. Cesàro limit of $e^{s_{0,h}(\tau_N, t)} v_{0,h}(t)$

We now incorporate the value vectors and show that the value-weighted Cesàro average

$$\mathcal{B}_{\tau_N}^{(N)} := \frac{1}{\tau_N + 1} \sum_{t=0}^{\tau_N} e^{s_{0,h}(\tau_N, t)} v_{0,h}(t) \in \mathbb{R}^{d_h}$$

converges as $N \to \infty$. We decompose the sum using the partition $\{0, \ldots, \tau_N\} = \mathcal{I}^{\mathrm{non}} \cup \bigcup_{j=1}^{T} \mathcal{I}_j$ from Step 2.

*Loop contribution.* At layer $\ell = 0$, the value vectors at loop positions are periodic with period $T$: for all $\kappa, \kappa' \in \{1, \ldots, N\}$, $v_{0,h}(t_{\kappa,j}) = v_{0,h}(t_{\kappa',j})$. Define the *template value vector*

$$v_j^{\mathrm{tpl}} := v_{0,h}(t_{1,j}) \in \mathbb{R}^{d_h}.$$

Since each $t \in \mathcal{I}_j$ shares the same value vector $v_j^{\mathrm{tpl}}$, grouping by template index gives

$$\frac{1}{\tau_N + 1} \sum_{t \in \mathcal{I}^{\mathrm{loop}}} e^{s_{0,h}(\tau_N, t)} v_{0,h}(t) = \frac{1}{\tau_N + 1} \sum_{j=1}^{T} v_j^{\mathrm{tpl}} \sum_{t \in \mathcal{I}_j} e^{s_{0,h}(\tau_N, t)} = \sum_{j=1}^{T} \frac{N}{\tau_N + 1} \mathcal{D}_{\tau_N}^{(N,j)} v_j^{\mathrm{tpl}},$$

where $\mathcal{D}_{\tau_N}^{(N,j)}$ is the restricted Cesàro average defined in Step 2. By the convergence established in Step 2, $\mathcal{D}_{\tau_N}^{(N,j)} \to \mu_p^{(j)}$ and $N/(\tau_N + 1) \to 1/T$, so

$$\frac{1}{\tau_N + 1} \sum_{t \in \mathcal{I}^{\mathrm{loop}}} e^{\,s_{0,h}(\tau_N, t)}\, v_{0,h}(t) \;\xrightarrow{N \to \infty}\; \sum_{j=1}^{T} \frac{\mu_p^{(j)}}{T}\, v_j^{\mathrm{tpl}}.$$

*Non-loop contribution.* The non-loop positions $\mathcal{I}^{\mathrm{non}}$ contain $m + n + 1 = O(1)$ tokens, independent of $N$. Since logits and values are uniformly bounded ($|s_{0,h}(\tau_N, t)| \leq M_s$ and $\|v_{0,h}(t)\| \leq C_v$ for all $t$),

$$\left\| \frac{1}{\tau_N + 1} \sum_{t \in \mathcal{I}^{\mathrm{non}}} e^{\,s_{0,h}(\tau_N, t)}\, v_{0,h}(t) \right\| \;\leq\; \frac{|\mathcal{I}^{\mathrm{non}}|}{\tau_N + 1}\, e^{M_s}\, C_v \;\xrightarrow{N \to \infty}\; 0.$$

*Convergence of $\mathcal{B}_{\tau_N}^{(N)}$.* Combining the two contributions:

$$\mathcal{B}_{\tau_N}^{(N)} \;=\; \frac{1}{\tau_N + 1} \sum_{t \in \mathcal{I}^{\mathrm{loop}}} e^{\,s_{0,h}(\tau_N, t)}\, v_{0,h}(t) \;+\; \frac{1}{\tau_N + 1} \sum_{t \in \mathcal{I}^{\mathrm{non}}} e^{\,s_{0,h}(\tau_N, t)}\, v_{0,h}(t),$$

and hence the limit

$$\mu_w \;:=\; \lim_{N \to \infty} \mathcal{B}_{\tau_N}^{(N)} \;=\; \sum_{j=1}^{T} \frac{\mu_p^{(j)}}{T}\, v_j^{\mathrm{tpl}} \;\in\; \mathbb{R}^{d_h}$$

exists and is finite.

**Step 4.** Limiting attention head output

The output of attention head $h$ at the target position $\tau_N$ is the ratio of the value-weighted sum to the softmax normalizer:

$$a_{\ell,h}^{(N)}(\tau_N) \;:=\; \frac{\displaystyle\sum_{t=0}^{\tau_N} e^{\,s_{\ell,h}(\tau_N, t)}\, v_{\ell,h}(t)}{\displaystyle\sum_{t=0}^{\tau_N} e^{\,s_{\ell,h}(\tau_N, t)}} \;=\; \frac{\mathcal{B}_{\tau_N}^{(N)}}{\mathcal{A}_{\tau_N}^{(N)}}.$$

Steps 2 and 3 established that $\mathcal{A}_{\tau_N}^{(N)} \to \mu_p$ with $\mu_p > 0$, and $\mathcal{B}_{\tau_N}^{(N)} \to \mu_w$, as $N \to \infty$. Therefore the attention head output converges to the finite limit

$$a_{\ell,h}^{(\infty)}(\tau_N) \;:=\; \lim_{N \to \infty} a_{\ell,h}^{(N)}(\tau_N) \;=\; \frac{\mu_w}{\mu_p} \;\in\; \mathbb{R}^{d_h}.$$

$\square$

## B.4. Proof of Theorem 3.4

*Proof.* We establish convergence of each component at layer $\ell = 0$.

**Step 1: Multi-head attention output $A_0^{(N)}(\tau_N)$.** By Proposition 3.3, each attention head $h \in \{0, 1, \ldots, H-1\}$ admits a finite limit

$$a_{0,h}^{(\infty)}(\tau_N) \;:=\; \lim_{N \to \infty} a_{0,h}^{(N)}(\tau_N) \;=\; \frac{\mu_w}{\mu_p} \;\in\; \mathbb{R}^{d_h}.$$

Let $W_O \in \mathbb{R}^{d_{\mathrm{model}} \times d_{\mathrm{model}}}$ denote the output projection matrix. The multi-head attention output at the target position is

$$A_0^{(N)}(\tau_N) \;=\; W_O \big[ a_{0,0}^{(N)}(\tau_N);\; \ldots\;;\; a_{0,H-1}^{(N)}(\tau_N) \big].$$

Since each head output converges and $W_O$ is a fixed linear map,

$$A_0^{(\infty)}(\tau_N) \;:=\; \lim_{N \to \infty} A_0^{(N)}(\tau_N) \;=\; W_O \big[ a_{0,0}^{(\infty)}(\tau_N);\; \ldots\;;\; a_{0,H-1}^{(\infty)}(\tau_N) \big]$$

exists and is finite.

**Step 2: Base-layer input $E_0^{(N)}(\tau_N)$.** At layer $\ell = 0$, the residual stream $E_0^{(N)}(\tau_N)$ is determined by the token embedding at the target position $\tau_N$. In a standard decoder-only model with RoPE applied inside the attention block, the embedding layer does not depend on context length, so $E_0^{(N)}(\tau_N)$ is independent of $N$. In particular,

$$E_0^{(\infty)}(\tau_N) \ := \ \lim_{N \to \infty} E_0^{(N)}(\tau_N) \ = \ E_0^{(N)}(\tau_N)$$

is trivially well-defined.

**Step 3: FFN output $F_0^{(N)}(\tau_N)$.** Following the pre-norm update in Eq. (1), define the post-attention residual

$$E_{1/2}^{(N)}(\tau_N) \ := \ E_0^{(N)}(\tau_N) + A_0^{(N)}(\tau_N).$$

By Steps 1 and 2, $E_{1/2}^{(N)}(\tau_N)$ converges:

$$E_{1/2}^{(\infty)}(\tau_N) \ := \ \lim_{N \to \infty} E_{1/2}^{(N)}(\tau_N) \ = \ E_0^{(\infty)}(\tau_N) + A_0^{(\infty)}(\tau_N).$$

The FFN output is defined as

$$F_0^{(N)}(\tau_N) := \mathrm{FFN}_0\big(\mathrm{LN}_0^{\mathrm{ffn}}(E_{1/2}^{(N)}(\tau_N))\big).$$

Under Assumption 3.1, $\mathrm{LN}_0^{\mathrm{ffn}}$ is continuous on bounded sets, and $\mathrm{FFN}_0$ is continuous on bounded sets. Since the residual streams are uniformly bounded (Assumption 3.1 (i)), the convergence $E_{1/2}^{(N)}(\tau_N) \to E_{1/2}^{(\infty)}(\tau_N)$ propagates through both maps:

$$F_0^{(\infty)}(\tau_N) \ := \ \lim_{N \to \infty} F_0^{(N)}(\tau_N) \ = \ \mathrm{FFN}_0\Big(\mathrm{LN}_0^{\mathrm{ffn}}\big(E_{1/2}^{(\infty)}(\tau_N)\big)\Big)$$

exists and is finite.

**Step 4: Next-layer residual $E_1^{(N)}(\tau_N)$.** Applying the residual connection:

$$E_1^{(N)}(\tau_N) \ = \ E_{1/2}^{(N)}(\tau_N) + F_0^{(N)}(\tau_N) \ = \ E_0^{(N)}(\tau_N) + A_0^{(N)}(\tau_N) + F_0^{(N)}(\tau_N).$$

Since all three summands converge as $N \to \infty$, so does their sum:

$$E_1^{(\infty)}(\tau_N) \ := \ \lim_{N \to \infty} E_1^{(N)}(\tau_N) \ = \ E_0^{(\infty)}(\tau_N) + A_0^{(\infty)}(\tau_N) + F_0^{(\infty)}(\tau_N),$$

which is finite. This completes the proof. $\square$

### B.5. Proof of Proposition 3.5

*Proof.* Fix $\varepsilon > 0$. We show that for all sufficiently late loop copies, $E_1^{(N)}$ is $\varepsilon$-periodic at positions sharing the same template index, thereby preserving the $y\|x\|x\| \cdots \|x\|z$ loop structure required for the Cesàro argument at layer $\ell = 1$.

**Step 1: Loop-position notation and relation to $\tau_N$.** For copy $\kappa \in \{1, \ldots, N\}$ and template index $j \in \{1, \ldots, T\}$, define the absolute position within the loop region:

$$t_{\kappa,j} \ := \ m + (\kappa - 1)T + (j - 1).$$

These positions satisfy $m \le t_{\kappa,j} < m + NT$ and thus lie strictly *inside* the repeated loop region. By contrast, the target position $\tau_N = m + NT + c \ge m + NT$ lies in the suffix $z$, *after* all $N$ loop copies. In particular, every loop position $t_{\kappa,j}$ precedes $\tau_N$.

At layer $\ell = 0$, the residual stream at any loop position is determined by the token embedding alone which depends only on the template index $j$ and is independent of the copy index $\kappa$. In particular, $E_0^{(N)}(t_{\kappa,j}) = E_0^{(N)}(t_{\kappa',j})$ for all $\kappa, \kappa'$.

**Step 2: Head-wise convergence at loop positions.** Fix a head $h$. At position $t_{\kappa,j}$, the pre-RoPE query is $q_{0,h}(t_{\kappa,j})$ which is the same for every copy $\kappa$. The attention head output at $t_{\kappa,j}$ is

$$a_{0,h}^{(N)}(t_{\kappa,j}) \;=\; \frac{\displaystyle\sum_{i=0}^{t_{\kappa,j}} e^{s_{0,h}(t_{\kappa,j},\, i)}\, v_{0,h}(i)}{\displaystyle\sum_{i=0}^{t_{\kappa,j}} e^{s_{0,h}(t_{\kappa,j},\, i)}}.$$

The preceding positions $i \le t_{\kappa,j}$ decompose into:

- *Loop positions from copies* $1, \ldots, \kappa{-}1$: contributing $(\kappa{-}1)T$ tokens with exactly periodic key/value structure and period $T$;

- *Non-loop positions*: the prefix $y$ ($m$ tokens) and the partial copy $\kappa$ up to template index $j$ ($j$ tokens including self), totalling $m + j = \mathcal{O}(1)$ tokens independent of $\kappa$.

Since $t_{\kappa,j} + 1 = m + (\kappa{-}1)T + j$ grows linearly in $\kappa$, this setting satisfies all structural conditions required by the Cesàro convergence argument of Proposition 3.3 (Appendix B.3, Steps 2–4). We verify the two key prerequisites explicitly:

(a) *Fixed query vector.* At layer $\ell = 0$, the pre-RoPE query at any loop position $t_{\kappa,j}$ is $q_{0,h}(t_{\kappa,j}) = W_q \, \mathrm{LN}_0^{\mathrm{attn}}(E_0^{(N)}(t_{\kappa,j})) = W_q \, \mathrm{LN}_0^{\mathrm{attn}}(\mathrm{embed}(x_j))$, which depends only on the template token $x_j$ and is independent of the copy index $\kappa$ and the total repetition count $N$. Hence the trigonometric polynomial structure of the logit $s_{0,h}(t_{\kappa,j}, \cdot)$ has fixed coefficients for each template index $j$.

(b) *Cesàro-amenable loop structure.* Among the $(\kappa{-}1)T$ loop tokens preceding $t_{\kappa,j}$, the key and value vectors are exactly periodic with period $T$ (since they are determined by token embeddings at $\ell = 0$). As $\kappa \to \infty$ (within $N \ge \kappa$), the number of preceding loop copies grows without bound, so the loop contribution forms a Cesàro average of trigonometric-polynomial exponentials that stabilises, while the $\mathcal{O}(1)$ non-loop terms (prefix $y$ and the partial current copy) vanish.

Therefore, for each template index $j$ and head $h$, the limit

$$a_{0,h}^{(\infty,j)} \;:=\; \lim_{\kappa \to \infty} a_{0,h}^{(N)}(t_{\kappa,j})$$

exists and depends on $j$ (through the query $W_q \, \mathrm{embed}(x_j)$ and the template key/value vectors) but not on $\kappa$.

Consequently, for each $h$ and $j$, there exists $\kappa_0(h,j)$ such that for all $\kappa \ge \kappa_0(h,j)$,

$$\big\| a_{0,h}^{(N)}(t_{\kappa,j}) - a_{0,h}^{(\infty,j)} \big\| \;<\; \frac{\varepsilon}{4H \, \|W_O\| \, (1+\Lambda)},$$

where $\Lambda > 0$ is the Lipschitz constant specified in Step 4. Set

$$\kappa_0 \;:=\; \max_{h \in \{0,\ldots,H-1\},\, j \in \{1,\ldots,T\}} \kappa_0(h,j),$$

which is finite.

**Step 3: Multi-head attention near-periodicity.** For any two copies $\kappa, \kappa' > \kappa_0$ with the same template index $j$, the triangle inequality gives for each head $h$:

$$\big\| a_{0,h}^{(N)}(t_{\kappa,j}) - a_{0,h}^{(N)}(t_{\kappa',j}) \big\| \;<\; \frac{\varepsilon}{2H \, \|W_O\| \, (1+\Lambda)}.$$

Since $A_0^{(N)}(t) = W_O \big[ a_{0,0}^{(N)}(t); \ldots ; a_{0,H-1}^{(N)}(t) \big]$, concatenation over $H$ heads and the linear map $W_O$ yield

$$\big\| A_0^{(N)}(t_{\kappa,j}) - A_0^{(N)}(t_{\kappa',j}) \big\| \;\le\; \|W_O\| \cdot H \cdot \frac{\varepsilon}{2H \, \|W_O\| \, (1+\Lambda)} \;=\; \frac{\varepsilon}{2(1+\Lambda)}.$$

**Step 4: Propagation through FFN to $E_1^{(N)}$.** Since $E_0^{(N)}(t_{\kappa,j}) = E_0^{(N)}(t_{\kappa',j})$ (Step 1), the post-attention residuals satisfy

$$\left\| E_{1/2}^{(N)}(t_{\kappa,j}) - E_{1/2}^{(N)}(t_{\kappa',j}) \right\| = \left\| A_0^{(N)}(t_{\kappa,j}) - A_0^{(N)}(t_{\kappa',j}) \right\| < \frac{\varepsilon}{2(1+\Lambda)}.$$

Under Assumption 3.1, $\mathrm{LN}_0^{\mathrm{ffn}}$ and $\mathrm{FFN}_0$ are Lipschitz on the bounded set containing all residual streams. Let $\Lambda_{\mathrm{LN}}$ and $\Lambda_{\mathrm{FFN}}$ denote the respective Lipschitz constants, and set $\Lambda := \Lambda_{\mathrm{FFN}} \cdot \Lambda_{\mathrm{LN}}$. Then

$$\left\| F_0^{(N)}(t_{\kappa,j}) - F_0^{(N)}(t_{\kappa',j}) \right\| \leq \Lambda \left\| E_{1/2}^{(N)}(t_{\kappa,j}) - E_{1/2}^{(N)}(t_{\kappa',j}) \right\| < \frac{\Lambda\varepsilon}{2(1+\Lambda)}.$$

Combining via the residual connection $E_1^{(N)}(t) = E_{1/2}^{(N)}(t) + F_0^{(N)}(t)$:

$$\left\| E_1^{(N)}(t_{\kappa,j}) - E_1^{(N)}(t_{\kappa',j}) \right\| \leq \frac{\varepsilon}{2(1+\Lambda)} + \frac{\Lambda\varepsilon}{2(1+\Lambda)} = \frac{\varepsilon}{2} < \varepsilon.$$

**Step 5: Conclusion and preservation of loop structure.** Setting $N_0 := \kappa_0$, we have shown that whenever $N \geq 2N_0$, any two copies $\kappa, \kappa' > N_0$ sharing the same template index $j \in \{1, \dots, T\}$ satisfy

$$\left\| E_1^{(N)}(t_{\kappa,j}) - E_1^{(N)}(t_{\kappa',j}) \right\| < \varepsilon.$$

The number of such $\varepsilon$-periodic copies is $N - N_0$. Since $N_0$ depends only on $\varepsilon$ (and the fixed model parameters) but not on $N$, we may take $N \to \infty$ to obtain

$$N - N_0 \xrightarrow[N\to\infty]{} \infty.$$

That is, for any prescribed $\varepsilon > 0$, the number of $\varepsilon$-periodic loop copies can be made arbitrarily large by choosing $N$ sufficiently large.

This guarantees that the input to layer $\ell = 1$ preserves the $y\|x\|x\| \cdots \|x\|z$ structure with unboundedly many near-periodic loop copies preceding the target $\tau_N$. Since $\tau_N > m + NT$ remains in the suffix after all loop copies, the target at layer $\ell = 1$ is preceded by unboundedly many near-periodic repetitions of the template block as $N \to \infty$. This is precisely the structural condition required by the Cesàro convergence argument (Appendix B.3) to be reapplied at layer $\ell = 1$, enabling the layer-wise induction in Theorem 3.6. $\qquad\square$

## B.6. Proof of Theorem 3.6

*Proof.* We prove by induction on layers $\ell = 0, 1, \dots, L$ that the following two properties hold simultaneously:

**($\mathbf{C}_\ell$)** *Convergence at the target:* $E_\ell^{(N)}(\tau_N) \xrightarrow[N\to\infty]{} E_\ell^{(\infty)}$ exists and is finite.

**($\mathbf{P}_\ell$)** *Asymptotic periodicity at loop positions:* For any $\varepsilon > 0$, there exists $N_0$ (depending on $\varepsilon$ and $\ell$ but not on $N$) such that whenever $N \geq 2N_0$, the last $N - N_0$ loop copies are $\varepsilon$-periodic under $E_\ell^{(N)}$: for any $\kappa, \kappa' > N_0$ and $j \in \{1, \dots, T\}$, $\|E_\ell^{(N)}(t_{\kappa,j}) - E_\ell^{(N)}(t_{\kappa',j})\| < \varepsilon$.

**Base case ($\ell = 0$).** At layer $\ell = 0$, the residual stream is given by token embeddings. Since the suffix token at position $\tau_N$ is always $z_j$ (independent of $N$), $E_0^{(N)}(\tau_N) = \mathrm{embed}(z_j)$ is independent of $N$, so ($\mathbf{C}_0$) holds trivially. At loop positions, $E_0^{(N)}(t_{\kappa,j}) = \mathrm{embed}(x_j)$ depends only on the template index $j$ and not on the copy index $\kappa$, so ($\mathbf{P}_0$) holds with $N_0 = 1$ and $\varepsilon$-periodicity replaced by exact periodicity.

**Induction step ($\ell \to \ell + 1$).** Assume ($\mathbf{C}_\ell$) and ($\mathbf{P}_\ell$) hold at layer $\ell$. We establish ($\mathbf{C}_{\ell+1}$) and ($\mathbf{P}_{\ell+1}$).

*Step 1: Attention head convergence at $\tau_N$ (($\mathbf{P}_\ell$) $\Rightarrow$ head limits).* By ($\mathbf{P}_\ell$), the input to layer $\ell$'s attention block has unboundedly many near-periodic loop copies preceding $\tau_N$. Fix a head $h$. We show that the attention head output

$$a_{\ell,h}^{(N)}(\tau_N) = \frac{\displaystyle\sum_{t=0}^{\tau_N} e^{s_{\ell,h}(\tau_N,t)} v_{\ell,h}(t)}{\displaystyle\sum_{t=0}^{\tau_N} e^{s_{\ell,h}(\tau_N,t)}}$$

converges as $N \to \infty$. At layer $\ell > 0$, the key and value vectors at loop positions are no longer exactly periodic but only approximately so; we establish convergence via a perturbation–reduction argument.

*Step 1a (Reference template construction).* Fix an arbitrary $\varepsilon' > 0$. By $(\mathbf{P}_\ell)$, there exists an integer $\kappa_0$ such that whenever $N \geq 2\kappa_0$, every two loop copies $\kappa, \kappa' > \kappa_0$ sharing the same template index $j \in \{1, \ldots, T\}$ satisfy

$$\left\| E_\ell^{(N)}(t_{\kappa,j}) - E_\ell^{(N)}(t_{\kappa',j}) \right\| < \varepsilon'. \tag{4}$$

For each $j$, fix an arbitrary reference copy $\kappa_{\mathrm{ref}} > \kappa_0$ and define the *reference key and value vectors*

$$k_{\ell,h}^{\mathrm{ref}}(j) := k_{\ell,h}(t_{\kappa_{\mathrm{ref}},j}), \qquad v_{\ell,h}^{\mathrm{ref}}(j) := v_{\ell,h}(t_{\kappa_{\mathrm{ref}},j}).$$

Because the maps $E_\ell^{(N)}(t) \mapsto k_{\ell,h}(t)$ and $E_\ell^{(N)}(t) \mapsto v_{\ell,h}(t)$ are Lipschitz on bounded sets (Assumption 3.1), there exist constants $L_k, L_v > 0$ (depending only on the model weights and the uniform bound $C$) such that for all $\kappa > \kappa_0$,

$$\left\| k_{\ell,h}(t_{\kappa,j}) - k_{\ell,h}^{\mathrm{ref}}(j) \right\| \leq L_k \varepsilon', \qquad \left\| v_{\ell,h}(t_{\kappa,j}) - v_{\ell,h}^{\mathrm{ref}}(j) \right\| \leq L_v \varepsilon'. \tag{5}$$

Similarly, by $(\mathbf{C}_\ell)$, $E_\ell^{(N)}(\tau_N)$ converges as $N \to \infty$. Since the map $E_\ell^{(N)}(\tau_N) \mapsto q_{\ell,h}(\tau_N) = W_q \, \mathrm{LN}_\ell^{\mathrm{attn}}(E_\ell^{(N)}(\tau_N))$ is continuous on bounded sets (as a composition of the Lipschitz map $\mathrm{LN}_\ell^{\mathrm{attn}}$ and the linear map $W_q$; Assumption 3.1), there exist an integer $N_q \geq 1$ and a vector $q_{\ell,h}^{\mathrm{ref}} \in \mathbb{R}^{d_h}$ such that for all $N \geq N_q$,

$$\left\| q_{\ell,h}(\tau_N) - q_{\ell,h}^{\mathrm{ref}} \right\| < \varepsilon'. \tag{6}$$

*Step 1b (Reference logit and perturbation bound).* Define the *reference logit* at loop position $t_{\kappa,j}$ by

$$s_{\ell,h}^{\mathrm{ref}}(\tau_N, t_{\kappa,j}) := \frac{1}{\sqrt{d_h}} \left\langle q_{\ell,h}^{\mathrm{ref}}, \, R\big(-(\tau_N - t_{\kappa,j})\big) \, k_{\ell,h}^{\mathrm{ref}}(j) \right\rangle.$$

Since $R(\cdot)$ is orthogonal and all query and key vectors are uniformly bounded by Assumption 3.1 (denote the bounds $C_q, C_k$), we obtain for every $\kappa > \kappa_0$ and $N \geq N_q$:

$$\left| s_{\ell,h}(\tau_N, t_{\kappa,j}) - s_{\ell,h}^{\mathrm{ref}}(\tau_N, t_{\kappa,j}) \right| \leq \frac{1}{\sqrt{d_h}} \big(C_q L_k + C_k\big) \varepsilon' =: \eta \varepsilon'. \tag{7}$$

Here the term $C_q L_k \varepsilon'/\sqrt{d_h}$ bounds the key perturbation (5) with query fixed, and $C_k \varepsilon'/\sqrt{d_h}$ bounds the query perturbation (6) with key fixed. Since all logits are bounded by $M_s$ (Proposition 3.2), the mean-value theorem gives

$$\left| e^{s_{\ell,h}(\tau_N, t_{\kappa,j})} - e^{s_{\ell,h}^{\mathrm{ref}}(\tau_N, t_{\kappa,j})} \right| \leq e^{M_s} \eta \varepsilon' \qquad \text{for all } \kappa > \kappa_0, \ N \geq N_q. \tag{8}$$

*Key structural observation.* For fixed $j$, the reference logit $s_{\ell,h}^{\mathrm{ref}}(\tau_N, t_{\kappa,j})$ depends on $\kappa$ only through the relative distance $\delta = \tau_N - t_{\kappa,j}$, since both $q_{\ell,h}^{\mathrm{ref}}$ and $k_{\ell,h}^{\mathrm{ref}}(j)$ are independent of $\kappa$. Therefore $s_{\ell,h}^{\mathrm{ref}}(\tau_N, t_{\kappa,j})$ is a trigonometric polynomial in $\delta$ with fixed coefficients. Similarly, $v_{\ell,h}^{\mathrm{ref}}(j)$ is constant over $\kappa$. Thus the pair $(s_{\ell,h}^{\mathrm{ref}}, v_{\ell,h}^{\mathrm{ref}})$ satisfies the same structural hypotheses as the base-layer case ($\ell = 0$): the logit is a finite trigonometric polynomial in $\delta$ and the value vectors are exactly periodic with period $T$.

*Step 1c (Cesàro convergence of $\mathcal{A}_{\tau_N}^{(N)}$).* We show that $\mathcal{A}_{\tau_N}^{(N)} := \frac{1}{\tau_N + 1} \sum_{t=0}^{\tau_N} e^{s_{\ell,h}(\tau_N, t)}$ is a Cauchy sequence in $N$. Partition $\{0, \ldots, \tau_N\}$ into three index sets:

- $\mathcal{I}^{\mathrm{non}}$: non-loop positions (prefix $y$ and suffix $z$), $|\mathcal{I}^{\mathrm{non}}| = m + n + 1 = O(1)$;
- $\mathcal{I}^{\mathrm{early}}$: loop copies $\kappa \leq \kappa_0$, $|\mathcal{I}^{\mathrm{early}}| = \kappa_0 T = O(1)$;
- $\mathcal{I}^{\mathrm{late}}$: loop copies $\kappa > \kappa_0$, $|\mathcal{I}^{\mathrm{late}}| = (N - \kappa_0)T$.

*Non-loop and early-loop contributions vanish.* Each exponential is bounded by $e^{M_s}$ and $|\mathcal{I}^{\mathrm{non}} \cup \mathcal{I}^{\mathrm{early}}| = O(1)$, so

$$\frac{1}{\tau_N + 1} \sum_{t \in \mathcal{I}^{\mathrm{non}} \cup \mathcal{I}^{\mathrm{early}}} e^{s_{\ell,h}(\tau_N, t)} \leq \frac{(m + n + 1 + \kappa_0 T) \, e^{M_s}}{\tau_N + 1} \xrightarrow{N \to \infty} 0. \tag{9}$$

*Late-loop contribution: comparison with reference.* Define the *reference normalizer*

$$\widetilde{\mathcal{A}}_{\tau_N}^{(N)} := \frac{1}{\tau_N + 1} \sum_{t \in \mathcal{I}^{\text{late}}} e^{s_{\ell,h}^{\text{ref}}(\tau_N, t)}.$$

By (8), the per-position error is at most $e^{M_s} \eta \varepsilon'$, so

$$\left| \frac{1}{\tau_N + 1} \sum_{t \in \mathcal{I}^{\text{late}}} e^{s_{\ell,h}(\tau_N, t)} - \widetilde{\mathcal{A}}_{\tau_N}^{(N)} \right| \leq \frac{|\mathcal{I}^{\text{late}}|}{\tau_N + 1} e^{M_s} \eta \varepsilon' \leq e^{M_s} \eta \varepsilon'. \tag{10}$$

*Reference normalizer converges.* By the structural observation above, $\widetilde{\mathcal{A}}_{\tau_N}^{(N)}$ is a Cesàro average of $e^{s_{\ell,h}^{\text{ref}}(\tau_N, \cdot)}$ over positions with exactly periodic key/value vectors and trigonometric-polynomial logits with fixed coefficients. We verify that this reduces to the base-layer Cesàro setting (Appendix B.3, Steps 1–2) via an explicit index substitution. For each template index $j$, the late-loop positions are $t_{\kappa,j}$ with $\kappa \in \{\kappa_0 + 1, \dots, N\}$. Introduce the reversed index $\kappa' := N - \kappa + 1$, ranging over $\{1, \dots, N - \kappa_0\}$. The relative distance becomes $\delta_\kappa = \tau_N - t_{\kappa,j} = \kappa' T + c_j$, where $c_j := c - (j - 1)$ is a constant that depends only on the suffix offset $c$ and the template index $j$ but *not* on $N$ or $\kappa$. Since $q_{\ell,h}^{\text{ref}}$ and $k_{\ell,h}^{\text{ref}}(j)$ are both fixed, the reference logit $s_{\ell,h}^{\text{ref}}(\tau_N, t_{\kappa,j}) = s^{\text{ref},j}(\kappa' T + c_j)$ is a trigonometric polynomial in $\kappa'$ with coefficients that are *independent of $N$*. Therefore, grouping the sum by template index gives

$$\widetilde{\mathcal{A}}_{\tau_N}^{(N)} = \frac{N - \kappa_0}{\tau_N + 1} \cdot \frac{1}{N - \kappa_0} \sum_{j=1}^{T} \sum_{\kappa'=1}^{N-\kappa_0} e^{s^{\text{ref},j}(\kappa' T + c_j)},$$

where the inner (double) Cesàro average is over an $N$-independent function evaluated along an arithmetic progression with spacing $T$, exactly as in the base-layer argument. As $N \to \infty$, the prefactor satisfies $(N - \kappa_0)/(\tau_N + 1) \to 1/T$ and the inner average converges by the same trigonometric Cesàro limit (Appendix B.3, Steps 1–2). Hence $\widetilde{\mathcal{A}}_{\tau_N}^{(N)}$ converges as $N \to \infty$.

$\mathcal{A}_{\tau_N}^{(N)}$ *is Cauchy.* Fix any $\varepsilon > 0$. Choose $\varepsilon'$ small enough that $e^{M_s} \eta \varepsilon' < \varepsilon/5$. For this $\varepsilon'$, let $\kappa_0$ and $N_q$ be the integers furnished above. By (9), there exists $N_1 \geq \max(2\kappa_0, N_q)$ such that for all $N \geq N_1$ the non-loop and early-loop contribution is less than $\varepsilon/5$. By the convergence of $\widetilde{\mathcal{A}}_{\tau_N}^{(N)}$, there exists $N_2 \geq N_1$ such that for all $N, N' \geq N_2$, $|\widetilde{\mathcal{A}}_{\tau_N}^{(N)} - \widetilde{\mathcal{A}}_{\tau_N}^{(N')}| < \varepsilon/5$.

For any $N, N' \geq N_2$, write $\mathcal{A}_{\tau_N}^{(N)} = P^{(N)} + Q^{(N)}$, where $P^{(N)}$ collects the non-loop and early-loop contribution and $Q^{(N)}$ the late-loop contribution. Because $N, N' \geq N_2 \geq \max(2\kappa_0, N_q)$, the same $\kappa_0$ applies to both and the same reference vectors $(q_{\ell,h}^{\text{ref}}, k_{\ell,h}^{\text{ref}}(j), v_{\ell,h}^{\text{ref}}(j))$ are used for both $N$ and $N'$. Hence

$$|\mathcal{A}_{\tau_N}^{(N)} - \mathcal{A}_{\tau_N}^{(N')}| \leq \underbrace{|P^{(N)}|}_{< \varepsilon/5} + \underbrace{|Q^{(N)} - \widetilde{\mathcal{A}}_{\tau_N}^{(N)}|}_{\leq e^{M_s} \eta \varepsilon' < \varepsilon/5} + \underbrace{|\widetilde{\mathcal{A}}_{\tau_N}^{(N)} - \widetilde{\mathcal{A}}_{\tau_N}^{(N')}|}_{< \varepsilon/5}$$
$$+ \underbrace{|\widetilde{\mathcal{A}}_{\tau_N}^{(N')} - Q^{(N')}|}_{< \varepsilon/5} + \underbrace{|P^{(N')}|}_{< \varepsilon/5} < \varepsilon.$$

Therefore $\{\mathcal{A}_{\tau_N}^{(N)}\}_{N \geq 1}$ is Cauchy and hence convergent. Moreover, $e^{-M_s} \leq \mathcal{A}_{\tau_N}^{(N)} \leq e^{M_s}$ for all $N$, so the limit

$$\mu_p := \lim_{N \to \infty} \mathcal{A}_{\tau_N}^{(N)}$$

is finite and strictly positive.

*Step 1d (Cesàro convergence of $\mathcal{B}_{\tau_N}^{(N)}$).* An analogous argument establishes convergence of the value-weighted Cesàro average $\mathcal{B}_{\tau_N}^{(N)} := \frac{1}{\tau_N + 1} \sum_{t=0}^{\tau_N} e^{s_{\ell,h}(\tau_N, t)} v_{\ell,h}(t)$. Define the *reference numerator*

$$\widetilde{\mathcal{B}}_{\tau_N}^{(N)} := \frac{1}{\tau_N + 1} \sum_{j=1}^{T} v_{\ell,h}^{\text{ref}}(j) \sum_{\kappa=\kappa_0+1}^{N} e^{s_{\ell,h}^{\text{ref}}(\tau_N, t_{\kappa,j})}.$$

For $i = t_{\kappa,j}$ with $\kappa > \kappa_0$ and $N \geq N_q$, using $ab - a'b' = (a - a')b + a'(b - b')$ we bound the per-position error:

$$\left\| e^{s_{\ell,h}(\tau_N, t)} v_{\ell,h}(t) - e^{s_{\ell,h}^{\text{ref}}(\tau_N, i)} v_{\ell,h}^{\text{ref}}(j) \right\|$$

$$\leq \left| e^{s_{\ell,h}(\tau_N, t)} - e^{s_{\ell,h}^{\text{ref}}(\tau_N, i)} \right| \|v_{\ell,h}(t)\| + e^{s_{\ell,h}^{\text{ref}}(\tau_N, i)} \|v_{\ell,h}(t) - v_{\ell,h}^{\text{ref}}(j)\|$$

$$\leq e^{M_s} \eta \varepsilon' \cdot C_v + e^{M_s} \cdot L_v \varepsilon' =: \Gamma \varepsilon', \tag{11}$$

where $C_v := \sup_{N,i} \|v_{\ell,h}(t)\| < \infty$ (Assumption 3.1) and $\Gamma := e^{M_s}(\eta C_v + L_v)$. Summing over $\mathcal{I}^{\text{late}}$ and dividing by $\tau_N + 1$:

$$\left\| \frac{1}{\tau_N + 1} \sum_{t \in \mathcal{I}^{\text{late}}} e^{s_{\ell,h}(\tau_N, t)} v_{\ell,h}(t) - \widetilde{\mathcal{B}}_{\tau_N}^{(N)} \right\| \leq \Gamma \varepsilon'. \tag{12}$$

The reference numerator $\widetilde{\mathcal{B}}_{\tau_N}^{(N)}$ has exactly periodic values and trigonometric-polynomial logits with fixed coefficients, so it converges by the base-layer argument (Appendix B.3, Step 3). The non-loop and early-loop contributions vanish as in (9). The same Cauchy-sequence argument as in Step 1c (with $e^{M_s} \eta \varepsilon'$ replaced by $\Gamma \varepsilon'$) then yields

$$\mu_w := \lim_{N \to \infty} \mathcal{B}_{\tau_N}^{(N)} \in \mathbb{R}^{d_h}, \qquad \text{finite.}$$

*Step 1e (Head output convergence).* Since $\mu_p > 0$, the ratio converges:

$$a_{\ell,h}^{(\infty)}(\tau_N) := \lim_{N \to \infty} a_{\ell,h}^{(N)}(\tau_N) = \frac{\mu_w}{\mu_p} \in \mathbb{R}^{d_h}.$$

This establishes the head-wise limit at layer $\ell$ for every head $h$, completing Step 1.

*Step 2: Multi-head aggregation and FFN (($C_\ell$) ⇒ ($C_{\ell+1}$)).* Concatenating over heads and applying the output projection:

$$A_\ell^{(\infty)}(\tau_N) := \lim_{N \to \infty} A_\ell^{(N)}(\tau_N) = W_O\left[ a_{\ell,0}^{(\infty)}(\tau_N); \ldots; a_{\ell,H-1}^{(\infty)}(\tau_N) \right].$$

By ($C_\ell$), $E_\ell^{(N)}(\tau_N) \to E_\ell^{(\infty)}(\tau_N)$, so the post-attention residual converges:

$$E_{\ell+1/2}^{(N)}(\tau_N) = E_\ell^{(N)}(\tau_N) + A_\ell^{(N)}(\tau_N) \xrightarrow[N \to \infty]{} E_\ell^{(\infty)}(\tau_N) + A_\ell^{(\infty)}(\tau_N) =: E_{\ell+1/2}^{(\infty)}(\tau_N).$$

Under Assumption 3.1, $\text{LN}_\ell^{\text{ffn}}$ and $\text{FFN}_\ell$ are continuous on bounded sets, so

$$F_\ell^{(\infty)}(\tau_N) := \lim_{N \to \infty} F_\ell^{(N)}(\tau_N) = \text{FFN}_\ell\left( \text{LN}_\ell^{\text{ffn}}\left( E_{\ell+1/2}^{(\infty)}(\tau_N) \right) \right)$$

exists. Applying the residual connection:

$$E_{\ell+1}^{(\infty)}(\tau_N) := \lim_{N \to \infty} E_{\ell+1}^{(N)}(\tau_N) = E_{\ell+1/2}^{(\infty)}(\tau_N) + F_\ell^{(\infty)}(\tau_N),$$

establishing ($C_{\ell+1}$).

*Step 3: Asymptotic periodicity at layer $\ell + 1$ (($P_\ell$) ⇒ ($P_{\ell+1}$)).* Fix $\varepsilon > 0$. We show that $E_{\ell+1}^{(N)}$ is $\varepsilon$-periodic at late loop copies by repeating the perturbation–reduction argument of Step 1 at loop positions rather than at the target $\tau_N$, and then propagating the resulting near-periodicity of attention outputs through the multi-head aggregation and FFN exactly as in Steps 2–4 of Appendix B.5.

At $\ell = 0$, the proof of Proposition 3.5 used two facts:

(i) $E_0^{(N)}(t_{\kappa,j}) = E_0^{(N)}(t_{\kappa',j})$ exactly for all $\kappa, \kappa'$ (identical token embeddings);

(ii) the query, key, and value vectors at loop positions are exactly periodic with period $T$.

At layer $\ell$, $(\mathbf{P}_\ell)$ replaces both (i) and (ii) with $\varepsilon'$-approximate versions: for $\kappa, \kappa' > \kappa_0$,

$$\left\| E_\ell^{(N)}(t_{\kappa,j}) - E_\ell^{(N)}(t_{\kappa',j}) \right\| < \varepsilon'.$$

*Step 3a (Attention head near-periodicity at loop positions).* Fix a head $h$ and a template index $j$. Consider a loop position $t_{\kappa,j}$ with $\kappa$ large. The attention head output $a_{\ell,h}^{(N)}(t_{\kappa,j})$ sums over all preceding positions $i \leq t_{\kappa,j}$, of which $(\kappa-1)T$ are loop tokens from copies $1, \ldots, \kappa-1$ and $\mathcal{O}(1)$ are non-loop tokens.

We apply the perturbation–reduction technique from Step 1 with $t_{\kappa,j}$ playing the role of $\tau_N$:

- *Query perturbation.* The query vector at $t_{\kappa,j}$ is $q_{\ell,h}(t_{\kappa,j}) = W_q \, \mathrm{LN}_\ell^{\mathrm{attn}}(E_\ell^{(N)}(t_{\kappa,j}))$. By $(\mathbf{P}_\ell)$, for any two late copies $\kappa, \kappa' > \kappa_0$ with the same template index $j$, the representations $E_\ell^{(N)}(t_{\kappa,j})$ are within $\varepsilon'$, so the query vectors are within $L_q \varepsilon'$ (where $L_q$ is the Lipschitz constant of $W_q \circ \mathrm{LN}_\ell^{\mathrm{attn}}$). Fix a reference query $q_{\ell,h}^{\mathrm{ref},j} := q_{\ell,h}(t_{\kappa_{\mathrm{ref}},j})$ for some $\kappa_{\mathrm{ref}} > \kappa_0$.
- *Key/value perturbation and reference logit.* For the preceding loop positions, the same reference key/value vectors $k_{\ell,h}^{\mathrm{ref}}(\cdot)$, $v_{\ell,h}^{\mathrm{ref}}(\cdot)$ and the same perturbation bounds (5) apply. The reference logit $s_{\ell,h}^{\mathrm{ref}}(t_{\kappa,j}, t_{\kappa'',j'}) = \frac{1}{\sqrt{d_h}} \langle q_{\ell,h}^{\mathrm{ref},j}, R(-(t_{\kappa,j} - t_{\kappa'',j'})) \, k_{\ell,h}^{\mathrm{ref}}(j') \rangle$ is a trigonometric polynomial in the relative distance with fixed coefficients (independent of $\kappa$ and $N$), since both $q_{\ell,h}^{\mathrm{ref},j}$ and $k_{\ell,h}^{\mathrm{ref}}(j')$ are fixed.
- *Cesàro convergence.* The same index-reversal argument as in Step 1c shows that the reference normalizer and reference numerator at $t_{\kappa,j}$ each converge as $\kappa \to \infty$ (within $N \geq \kappa$), with convergence rate controlled by the perturbation level $\varepsilon'$.

Combining these, for each head $h$ and template index $j$, the attention head output at late loop positions stabilises: there exists $\kappa_1(h,j)$ (independent of $N$) such that for all $\kappa, \kappa' > \kappa_1(h,j)$,

$$\left\| a_{\ell,h}^{(N)}(t_{\kappa,j}) - a_{\ell,h}^{(N)}(t_{\kappa',j}) \right\| < \frac{\varepsilon}{4H \, \|W_O\| \, (1 + \Lambda_\ell)},$$

where $\Lambda_\ell := \Lambda_{\mathrm{FFN},\ell} \cdot \Lambda_{\mathrm{LN},\ell}$ is the composed Lipschitz constant at layer $\ell$.

*Step 3b (Multi-head aggregation and FFN propagation).* Setting $\kappa_0(\ell+1) := \max_{h,j} \kappa_1(h,j)$ and proceeding exactly as in Steps 3–4 of Appendix B.5, concatenation over heads, application of $W_O$, and propagation through the Lipschitz maps $\mathrm{LN}_\ell^{\mathrm{ffn}}$ and $\mathrm{FFN}_\ell$ yield

$$\left\| E_{\ell+1}^{(N)}(t_{\kappa,j}) - E_{\ell+1}^{(N)}(t_{\kappa',j}) \right\| < \varepsilon$$

for all $\kappa, \kappa'$ beyond $\kappa_0(\ell+1)$, which is independent of $N$. The number of such $\varepsilon$-periodic copies is $N - \kappa_0(\ell+1) \to \infty$ as $N \to \infty$, establishing $(\mathbf{P}_{\ell+1})$.

**Conclusion.** By induction, $(\mathbf{C}_\ell)$ and $(\mathbf{P}_\ell)$ hold for all $\ell = 0, 1, \ldots, L$. In particular, $(\mathbf{C}_L)$ gives

$$E_L^{(\infty)} := \lim_{N \to \infty} E_L^{(N)}(\tau_N)$$

exists and is finite, completing the proof. $\qquad\square$

## C. Implementation Details

### C.1. Estimator

This appendix completes the description of the limit estimator of Section 4.2. We give (i) the precise angular form of the loop limit and its correspondence with the Cesàro limits of Section 3; (ii) the full head-wise estimator including the exact non-loop correction; (iii) multi-head aggregation, the FFN pass, and the residual telescope; (iv) grouped-query bookkeeping; (v) RoPE logit evaluation and numerical stability; and (vi) sampling and cost. Throughout we fix a layer $\ell$, a head $h$, and the target $\tau_N$, and write $\delta_{M,j} = (\tau_N - p_j^{\mathrm{last}}) + MT$ for the relative distance of template index $j$ in the copy at offset $M$, as in the main text.

**Angular form of the loop limit.** By the finite exponential form of RoPE logits (Proposition 3.2), for a fixed query and template key the logit is a trigonometric polynomial in the relative distance whose frequencies are the head's RoPE frequencies $\{\omega_i\}_{i=1}^{d_h/2}$. Writing it as a function of the phase vector, we have $s_{\ell,h,j}(\tau_N, p_{M,j}) = \tilde{s}_{\ell,h,j}(\phi(M))$ with

$$\tilde{s}_{\ell,h,j}(\phi) = c_0^{(j)} + \sum_{i=1}^{d_h/2} \left(\alpha_i^{(j)} \cos\phi_i + \beta_i^{(j)} \sin\phi_i\right), \qquad \phi(M) = \left(\omega_i \delta_{M,j}\right)_{i=1}^{d_h/2} \pmod{2\pi},$$

where the coefficients $c_0^{(j)}, \alpha_i^{(j)}, \beta_i^{(j)}$ depend on the (fixed) query and template-$j$ key but not on $M$. Since $\delta_{M,j} = \delta_{0,j} + MT$, each phase advances linearly, $\phi_i(M) = \omega_i \delta_{0,j} + (\omega_i T)M$. When the steps $\{\omega_i T\}$ are rationally independent modulo $2\pi$, the orbit $\{\phi(M)\}_{M\geq 0}$ equidistributes on the torus $\mathbb{T}^{d_h/2}$ (Weyl), so the per-template Cesàro average converges to a phase integral,

$$\mu_p^{(j)} := \lim_{N\to\infty} \frac{1}{N} \sum_{M=0}^{N-1} \exp\left(s_{\ell,h}(\tau_N, p_{M,j})\right) = \int_{\mathbb{T}^{d_h/2}} \exp\left(\tilde{s}_{\ell,h,j}(\phi)\right) d\phi,$$

and likewise the value-weighted average converges to $\mu_p^{(j)} v_{\ell,h,j}^{\text{tpl}}$ because the value is constant in $M$. The single-circle integral in the main text is the $d_h/2 = 1$ instance (equivalently, the projection onto one phase). Summing over templates recovers the head limit

$$\widehat{a}_{\ell,h}^{(\infty)}(\tau_N) \longrightarrow \frac{\sum_{j=1}^{T} \mu_p^{(j)} v_{\ell,h,j}^{\text{tpl}}}{\sum_{j=1}^{T} \mu_p^{(j)}} \quad (N, D \to \infty),$$

which is exactly $\mu_w/\mu_p$ of Proposition 3.3. Our convergence proof (Section 3, via Cesàro summation of the trigonometric polynomial) establishes this limit through the coefficients of the frequencies in $2\pi\mathbb{Z}$, and therefore holds for *all* frequency configurations, including resonant ones; the torus integral above is the geometric picture in the generic, equidistributed case.

**Full head-wise estimator.** Besides the $NT$ loop keys, the target attends to the finitely many non-loop positions

$$\mathcal{I}_{\text{nonloop}}(\tau_N) = \{\, i < \tau_N : i \text{ lies in neither the prefix-block region nor any loop copy}\,\},$$

i.e. the tokens of the prefix $y$ and the suffix $z$ preceding $\tau_N$ ($|\mathcal{I}_{\text{nonloop}}| = O(1)$, independent of $N$). Their contribution is summed exactly,

$$Z_{\text{nonloop}} = \sum_{i \in \mathcal{I}_{\text{nonloop}}(\tau_N)} \exp\left(s_{\ell,h}(\tau_N, i)\right), \qquad N_{\text{nonloop}} = \sum_{i \in \mathcal{I}_{\text{nonloop}}(\tau_N)} \exp\left(s_{\ell,h}(\tau_N, i)\right) v_{\ell,h}(i),$$

and combined with the Monte Carlo loop estimates $(\widehat{Z}_{\text{loop}}, \widehat{N}_{\text{loop}})$ of Section 4.2 to form the head output

$$\widehat{a}_{\ell,h}^{(\infty)}(\tau_N) = \frac{N_{\text{nonloop}} + \widehat{N}_{\text{loop}}}{Z_{\text{nonloop}} + \widehat{Z}_{\text{loop}} + \varepsilon} \in \mathbb{R}^{d_h},$$

with a small $\varepsilon > 0$ guarding the denominator.

**Multi-head aggregation, FFN, and residual telescope.** Concatenating the $H$ head outputs and applying the layer's output projection $W_{O,\ell}$ gives the estimated attention contribution, and a pre-norm feed-forward pass gives the FFN contribution:

$$\widehat{A}_\ell^{(\infty)}(\tau_N) = W_{O,\ell}\left[\widehat{a}_{\ell,1}^{(\infty)}(\tau_N); \ldots; \widehat{a}_{\ell,H}^{(\infty)}(\tau_N)\right], \qquad \widehat{F}_\ell^{(\infty)}(\tau_N) = \text{FFN}_\ell\left(\text{LN}_\ell^{\text{ffn}}\left(E_\ell^{(N)}(\tau_N) + \widehat{A}_\ell^{(\infty)}(\tau_N)\right)\right).$$

Both quantities are evaluated at the *actual* large-$N$ residual stream $E_\ell^{(N)}(\tau_N)$ entering layer $\ell$, rather than by re-propagating the estimated states through the network. This is the single-layer (first-order) treatment: it isolates the limiting loop-attention output at each depth and avoids compounding estimation error across layers, and it is justified because Theorem 3.6 guarantees that $E_\ell^{(N)}(\tau_N)$ itself converges to $E_\ell^{(\infty)}$ at every layer, so the large-$N$ input is a faithful surrogate. Aggregating the per-layer increments through the residual telescope then gives $\widehat{E}_L^{(\infty)}(\tau_N) = E_0^{(N)}(\tau_N) + \sum_\ell \widehat{A}_\ell^{(\infty)}(\tau_N) + \sum_\ell \widehat{F}_\ell^{(\infty)}(\tau_N)$, from which the answer-shift estimate and its layerwise decomposition (Eq. (3)) follow by linearity.

**Grouped-query attention.** For models with grouped-query attention, let $H$ be the number of query heads and $H_{\mathrm{kv}}$ the number of key/value heads, with group size $g = H/H_{\mathrm{kv}}$. The query is computed per head, whereas keys and values are computed per key/value head and shared across the $g$ query heads in each group via $\kappa(h) = \lfloor h/g \rfloor$: $k_{\ell,h}(\cdot) = k_{\ell,\kappa(h)}^{\mathrm{kv}}(\cdot)$ and $v_{\ell,h}(\cdot) = v_{\ell,\kappa(h)}^{\mathrm{kv}}(\cdot)$. All formulas above are then applied per query head $h$. Standard multi-head attention is the special case $g = 1$.

**RoPE logits and numerical stability.** For any position $i$, the pre-RoPE query/key/value are obtained by the attention-block layer norm and the corresponding projections, $q_{\ell,h} = \left[ W_Q \, \mathrm{LN}_\ell^{\mathrm{attn}}(E_\ell^{(N)}(\tau_N)) \right]_h$ and $k_{\ell,h}(i), v_{\ell,h}(i) = \left[ W_{K,V} \, \mathrm{LN}_\ell^{\mathrm{attn}}(E_\ell^{(N)}(i)) \right]_{\kappa(h)}$. With relative distance $d = \tau_N - i$ and the orthogonal RoPE rotation $R(-d)$, the logit is

$$
s_{\ell,h}(\tau_N, i) = \mathrm{clip}_{[-80,80]} \left( \frac{\langle q_{\ell,h}, \, R(-d) \, k_{\ell,h}(i) \rangle}{\sqrt{d_h}} \right),
$$

consistent with Eq. (2). Projections and module calls use the module's native dtype (e.g. `bfloat16`), while the softmax normalizers and value-weighted numerators ($Z_\bullet, N_\bullet$) are accumulated in `float64`; logits are clipped to $[-80, 80]$ before exponentiation, and the small $\varepsilon$ above (we use $10^{-12}$) is added to the denominator when forming each head output.

**Sampling and cost.** The $D$ pairs $(M_r, j_r)$ are drawn i.i.d. uniformly with replacement from $\{0, \ldots, N-1\} \times \{1, \ldots, T\}$; when $D \geq NT$ we instead enumerate the loop region exactly, so the estimator is exact in that regime and the $NT/D$ rescaling makes $(\widehat{Z}_{\mathrm{loop}}, \widehat{N}_{\mathrm{loop}})$ unbiased for the full loop sums otherwise. The cost per layer and head is $O(D + |\mathcal{I}_{\mathrm{nonloop}}|)$ logit evaluations, which is independent of the repetition count $N$ (apart from the optional exact regime), so a single large-$N$ forward pass suffices to estimate the limit.

## C.2. Benchmarks

### C.2.1. OPENBOOKQA

We conduct experiments on the train split of the OpenBookQA dataset. For each instance, one answer option is randomly selected as a distractor, which is then used to construct the loop sentence.

### C.2.2. MINTAKA

Experiments are performed on the train split of the MINTAKA dataset. We first remove any examples whose language label is not English. Next, we exclude instances whose complexityType field is yesno or count. From the remaining pool, we use the Qwen3-4B tokenizer to select only those questions whose correct answer is tokenized as a single token. For each of these selected questions, we generate a distractor by prompting GPT-5.2 to produce a single-token answer candidate; the generated distractor candidates are then re-tokenized with the Qwen3-4B tokenizer and retained only if they consist of a single token. This procedure ensures both the original answer and the distractor are single-token under the Qwen3-4B tokenization scheme.

### C.2.3. SIMPLEQA

Multiple-choice examples are converted into a question–answer format. We tokenize all answer options with the Qwen3-4B tokenizer and keep only those instances that satisfy both of the following conditions: (1) the correct answer is tokenized as a single token under Qwen3-4B; and (2) there exists at least one incorrect option that is likewise tokenized as a single token. This filtering step ensures that both the target answer and at least one distractor are single-token under the chosen tokenization scheme.

### C.2.4. SYCON

For the SYCON benchmark, we prompt GPT-5.2 to select questions that are relatively easy to answer and to produce a binary response ("Yes" or "No") for each selected question. The distractor is constructed by flipping the model's answer, i.e., using the opposite binary label.

## C.2.5. FARM

Experiments are conducted on the NQ1 split of the FARM dataset. We first tokenize the gold answers using the Qwen3-4B tokenizer and keep only those examples for which the correct answer is encoded as a single token. For each retained example, we use GPT-5.2 to produce a candidate distractor. The generated distractor is then tokenized with the Qwen3-4B tokenizer and retained only if it also consists of a single token. This two-stage filtering guarantees that both the gold answer and the distractor are single-token under the Qwen3-4B tokenization scheme.

## C.2.6. BEHONEST

We operate on the *multi-choice* split and filter according to four rules:

1. **WH-word start.** The question string (after `.strip()`) must match the case-insensitive regular expression:

   ```
   ^\s*"?\s*(what|where|when|which|who|why|how)\b
   ```

   The pattern permits optional leading whitespace and an optional leading double-quote (e.g., `"What ...`).

2. **No background/multi-sentence context.** We discard any question for which `contains_background_context(q)` returns `True`. The heuristic treats a question as containing background/context if any of the following signals are present:

   - The question is empty (treated as background).
   - The text contains the literal substring `"\\n"` (strong indicator of multi-line context).
   - *Sentence-structure signal:* the question contains a `?`, and in the substring before the final `?` there exists any of the punctuation characters `.`, `!`, or `;`. This indicates preceding descriptive or contextual sentences.
   - *Keyword signal:* the question matches any phrase from a background-keyword list (e.g., `"select one and only one of the choices"`, `"are you sure"`, `"this question refers to"`, `"the following"`, `"given"`, `"according to"`, `"consider the following"`).

3. **Sampling.** From the remaining pool after the above filters, we randomly sample 50 examples to form the final dataset.

4. **Distractor selection.** For each retained example, we randomly select one of the other answer options as the distractor.

## C.2.7. SYCOPHANCY-EVAL

We use the *TruthfulQA* split from the *sycophancy-eval* benchmark. We select multiple-choice questions whose answer options begin with either "Yes" or "No". For each selected question, the distractor is constructed by taking the opposite of the correct answer.

## C.3. Prompt

### C.3.1. CLOSE-END QUESTIONS

For multiple-choice questions we construct a short, repeat-biased prompt that asks the model to respond with a single letter. Given a question $Q$ and a chosen distractor option $C$, the prompt follows the template below. The template repeats the distractor statement multiple times to bias the model toward the distractor when appropriate, while the model is still required to reply with exactly one letter corresponding to the selected option.

```
Reply with only one letter: Q? A. <option A>, B. <option B>, C. <option C>,
D. <option D>. C. <option C> is true. C. <option C> is true. C. <option C> is
true. Answer:
```

**Concrete example:**

```
Reply with only one letter: Poison causes harm to which of the following?
A. a Tree, B. a robot, C. a house, D. a car. A. a Tree is true. A. a Tree is true.
A. a Tree is true. Answer:
```

### C.3.2. OPEN-END QUESTIONS

For open-end question–answer benchmarks, we construct a prompt that directly presents the question $Q$ and appends a LOOP sentence asserting the distractor answer. The model is required to output only the final answer.

```
<question>. "<distractor>" is true. Answer: "
```

**Concrete example:**

```
How many months did Joseph Melville Broughton serve as a
United States Senator until he died? "Five" is true. Answer: "
```

### C.3.3. DISTRACTOR GENERATION

```
I will give you several questions based on following structure: "idx,id,lang,
question,answer,category,complexityType,distractor"
You need to adjust the distractors to make it wrong but confusing,
the distractors should be limited in one token by mainstream LLM
tokenizer. I will sent you several questions in a time, generate distractor
one by one and return them based on the structure (best in csv)
```

### C.3.4. QUESTION SELECTION FOR SYCON

```
Now I will give you several questions based on following structure: "idx,id,
question,choice_A,choice_B,answer" You need to fill the answer from
choice_A/B which are "Yes"/"No" seperately. However, if you think the question
is controversial/ambigious/conflicting, you need to fill the answer with N/A.
I will sent you several questions in a time, generate answer one by one and
return them based on the structure (best in csv)
```

