# OpenReview forum: "Not All Answers Are Contextually Persuadable: Inference Dynamics in Large Language Models under Contextual Influence"
_ICML.cc/2026/Conference — ICML 2026 regular_

### Official Review · Reviewer_ZBV4 · 2026-02-23

**Soundness:** 3
**Presentation:** 3
**Significance:** 3
**Originality:** 3
**Overall Recommendation:** 3
**Confidence:** 3

**Summary:**

This paper discusses the impact of repetition in a prompt through a theoretical lens. It argues that repetition is not a factor for ICL and could (or could not, depending on the situation) hence influence a model's output. Empirical results to back up this assertion are provided.

**Compliance With Llm Reviewing Policy:**

Affirmed.

**Key Questions For Authors:**

1. What is the impact of assumption 3.2 (Lipschitzness and continuity) on your results? Assuming Lipschitz is pretty standard on the field because it makes the maths easier, but it does not fully adjust to realistic implementations. Same goes for continuity.
2. Would repetition sensitivity/lack thereof be more related to finetuning and not to your findings? Alternatively, your theoretical results appear to suggest it to be an intrinsic property of the transformer, not of an aligned LLM. But in that case, wouldn't that be observable in an untrained transformer?
3. How would this compare to other approaches which could lead to the same result, such as circuit-based analysis?

**Limitations:**

No -- see questions, strengths and weaknesses above

**Strengths And Weaknesses:**

This is a very strong paper, but needs improvement. The main points of improvement are related to framing, writing, and discussion.

It is solid from a technical perspective, but the assumptions are a bit strong and the experiments are related to pretrained LLMs, thus there is no way to distinguish between the effects of pretraining (and, by consequence, ICL), versus the impact of the assumptions in the theory.

I've added some suggestions for improvement at the end of this review.

## Writing:
1. It is a bit grandiose to refer to your results as 'surprising'. I recommend more objective writing throughout, especially since this bit is repeated in the abstract and the intro.
  - Likewise, the related works section is somewhat biased towards very new works, and does not account for fundamental work in this field. This, in turn, harms your framing. For example, (L88, C2, p2) 'While such sensitivity underlies effective prompting and in-context learning, it also raises concerns about prediction consistency and robustness under spurious or misleading contextual signals'. This is the core motivation of your work, albeit these concerns have been known since at least 2017 with Jia and Liang.
2. The citations from P1. C2 L28 aren't really needed given the context.
3. The intro is quite vague. '(...) it captures a fundamental primitive underlying many prompt-based steering and persuasion techniques'. This is unclear, and is accompanied with citations that do not appear to be related to the claims (otherwise, maybe expalin why are they there?).

## Framing
Going back to the framing, wouldn't repeated assertions / sensitivity to prompting / ICL / etc be simply a product of alignment and pretraining? For this to be a fully independent result, an unfinetuned transformer should present the same results. It is dubious it would, and thus the interplay between your results and pretraining is a lot more nuanced.
This is particularly important given that theoretical results are meant to be generalisable; and assumptions on alignment are not discussed either. Nonetheless, the motivation is mostly around the impact of prompting on pretrained models (re my point on Jia and Liang above).
Don't get me wrong: this is a good paper, but the framing needs work to make it a solid contribution.

## Discussion
- The assumptions on Lipschitzness and continuity, although common and having the property of making maths extremely easy, needs to be carefully discussed here. There's no such discussion--nor empirical results on corner cases, as per above--and thus it is quite probable that your analysis is more impacted by these results and the empirical observations are not fully confirmatory of your work.
- Statistical significance analyses are strongly required when making in the limit claims. I might have missed these, and I appreciate figure 2, but if they are not there I highly encourage the authors to include these to back their findings.

## Minor notes:
- It is still somewhat unclear from the intro/Figure 1 what repetition is in-context. I recommend simplifying that picture to a single example which illustrates your framing more clearly.
- Parameter calls for the LLMs need to be disclosed (though I might have missed this one)
- Watch out for your citations: lots of 'llms' and malformed caps. Other older papers have already been accepted to venues (e.g., Barbero et al was from last year's ICLR) so the citations need updating.
- Other approaches to ICL analysis need to be discussed here. Convexity-based approaches--like yours--aren't the only ones: circuit-based and learning-theory-based analyses could also be used for this analysis _without_ being exposed to the issues around pretraining I mentioned.

---

> ### Author Rebuttal · Authors · 2026-03-31
>
> We thank the reviewer for the constructive feedback and will revise the paper to improve presentation.
>
> ---
>
> > **Q1: Impact of Lipschitzness and continuity**
>
> Lipschitzness and continuity are technical regularity conditions ensuring bounded residual dynamics, which are needed in Theorems 3.5 and 3.6 to propagate convergence across layers. Under this assumption, normalized hidden states remain in a fixed bounded set; together with continuity of the attention and FFN blocks, this implies bounded residual updates and hence bounded layerwise residual streams.
>
> **Exp 1. Layerwise boundedness across depth**
>
> We assess this assumption empirically by computing the bound induced by Lipschitzness and continuity. We use $C_0+\\sum_{j=0}^{L-1}(C_{A,j})$, where $C_0, C_{A,j}$ are the bound for the input and the attention residual state. **Table 1** shows that this gives a tight upper bound, supporting that the residual stream remains well bounded in practice.
>
> **Table 1: Tightness and coverage of predicted bound**
> |Layers|Falcon3-3B Ratio|Falcon3-3B Cov|Qwen3-4B Ratio|Qwen3-4B Cov|Mistral-7B Ratio|Mistral-7B Cov|
> |---|---:|---:|---:|---:|---:|---:|
> |Front|0.27|1.00|0.46|1.00|0.28|1.00|
> |Mid|0.07|1.00|0.24|1.00|0.09|1.00|
> |Back|0.03|1.00|0.06|1.00|0.04|1.00|
> |**Mean**|**0.13**|**1.00**|**0.26**|**1.00**|**0.13**|**1.00**|
>
> ---
>
> > **Q2: Convergence in untrained transformer**
>
> **Exp 2: Layer-wise convergence in randomly initialized models**
>
> We test randomly initialized models to separate architectural effects from training effects. **Table 2** shows a high convergence rate and only slightly below the fine-tuned regime, indicating the intrinsic property of the transformer.
>
> **Table 2: Layer-wise convergence of randomly initialized models**
> |Model|α=0.1(A)|α=0.1(F)|α=0.05(A)|α=0.05(F)|
> |---|---:|---:|---:|---:|
> |Falcon3-3B|98.7|98.8|95.6|96.5|
> |Qwen3-4B|95.0|95.3|93.4|94.7|
> |Mistral-7B|97.3|98.2|95.6|96.2|
>
> ---
>
> > **Q3: Circuit-based analysis**
>
> Our analysis is complementary to circuit-based analysis. Circuit-based analysis can localize and causally validate the layers driving repetition-induced shifts, but it does not by itself characterize the global trajectory or asymptotic limit of inference under repetition. Our approach provides that global view, while circuit-based analysis explains where the effect is implemented. To address the reviewer’s question, we conduct three circuit-oriented experiments.
>
> **Exp 3. Layer-wise localization of repetition-induced shifts**
>
> We perform layer-wise analysis to localize layers in charge of repetition-induced answer shifts. For each layer, we compare its contribution in Eq. (3) with the final shift **Table 3**. The effect is often concentrated in a few late attention layers, which show strong sign alignment and large relative contribution, while FFN components are much less consistently aligned.
>
> **Table 3: Top Layers by Sign Alignment and Contribution**
> |Model|Layer|Sign(A)|Ratio(A)|Sign(F)|Ratio(F)|
> |---|---|---:|---:|---:|---:|
> |Falcon3-3B|20|99%|1.60|53%|1.57|
> ||21|68%|0.79|47%|1.06|
> |Qwen3-4B|35|98%|2.34|3%|1.02|
> ||34|77%|1.00|28%|0.85|
> |Mistral-7B|30|98%|1.40|45%|0.72|
> ||31|91%|1.00|5%|2.30|
>
> **Exp 4. Causal intervention on dominant attention layers**
>
> We perform a causal test via activation intervention. At N=1000, we replace the residual stream at the top-1/top-2 attention layers with the corresponding clean stream from N=0, and recompute output probabilities. **Table 4** shows consistent increases in  $P_{correct}$ and decreases in $P_{wrong}$​, indicating that these late attention layers causally mediate the repetition-induced shift toward the wrong answer.
>
> **Table 4: Causal Test via Attention Transplant**
> |Model|Transplant|$ΔP_{correct}/P_{correct,1000}$|sig.|$ΔP_{wrong}/P_{wrong,1000}$|sig.|
> |---|---|---:|---:|---:|---:|
> |Falcon3-3B|Top-1|+0.53|***|-0.39|***|
> ||Top-2|+6.18|***|-0.64|**|
> |Qwen3-4B|Top-1|+1.28|***|-0.06|***|
> ||Top-2|+0.40|**|-0.06|***|
> |Mistral-7B|Top-1|+0.36|***|-0.24|***|
> ||Top-2|+1.26|***|-0.83|***|
>
> **Exp 5. Cross-model variation in answer recovery**
>
> We examine answer recovery after intervention by testing whether intervening on the dominant attention layers recovers the original answer on flipped samples. **Table 5** shows heterogeneous recovery across models, with substantial recovery for Falcon3-3B but limited recovery for others.
>
> **Table 5: Recovery on Flipped Samples**
> |Model|Transplant|Recovery|
> |---|---|---:|
> |Falcon3-3B|Top-1|23.5%|
> ||Top-2|54.9%|
> |Qwen3-4B|Top-1|4.5%|
> ||Top-2|6.0%|
> |Mistral-7B|Top-1|0.0%|
> ||Top-2|7.7%|
>
> These results show that circuit-based analysis complements our framework by localizing and causally validating the responsible layers, while our approach captures the global and asymptotic behavior under repetition.
>
> ---
>
> > **Minor notes**
>
> We thank the reviewer for the informative feedback and update our Figure description and related work accordingly.

---

> > ### Author Rebuttal · Reviewer_ZBV4 · 2026-04-03
> >
> > Thank you for your response! I'm satisfied with the (thorough) responses around Lipschitzness, but my concerns around writing and framing remain.

---

### Official Review · Reviewer_JMWh · 2026-03-06

**Soundness:** 3
**Presentation:** 3
**Significance:** 2
**Originality:** 2
**Overall Recommendation:** 4
**Confidence:** 3

**Summary:**

The paper studies Transformer inference dynamics in the setting of contextual repetition: when multiple repetitions of a single suggestive answer are appended after a question, to steer the model towards that answer. The core result (Theorem 3.1) is that the Transformer residual streams after each layer converge as the number of repetitions goes to infinity. The paper further proposes an answer-shift metric and estimator for its asymptotic limit.
Experiments in QA tasks confirm the theorized layer-wise stability, and observed answer changes show good alignment with answer-shift estimates.

**Compliance With Llm Reviewing Policy:**

Affirmed.

**Final Justification:**

The rebuttal as been very engaging and the authors provided a lot more context to the Bayesian connection. However, I believe that the paper's strong claim “repeated contextual assertions do not act as accumulating evidence during inference” (as stated in the abstract) is not fully justified, but affects the narrative and positioning of the paper. Regardless, I find the results interesting and worthy to share already (hence weak accept), but believe that the paper could still be significantly improved.

**Key Questions For Authors:**

**Questions / Suggestions**

Q1. The authors only consider fixed repetition counts 500, 750 and 1000. It would be interesting to report, at least for the case study examples, when an answer flip occurs (if there is one), and whether it is correlated with model size.

Q2. The theoretical analysis heavily exploits elements of the Transformer architecture, such as attention and ROPE. While these are very reasonable choices, I wonder how much the approach would transfer to other architectures. Empirically, do other architectures show similar stabilization of answer flips at similar repetition counts?

**Smaller comments**

C1. What does Figure 2 show exactly? x and y labels are missing, and the caption does not give sufficient description.

C2.  It is confusing what exactly is reported on the x axis of Figure 3. Section 5.3 states that KL divergences are computed between the model’s predictive distributions at pairs of repetition lengths. The plot shows Kl trajectories as a function of the repetition length. Does this mean that the reference distribution for the KL is the predictive distribution right before the first answer was appended to the prompt?

**Limitations:**

No explicit limitations of the work are discussed.

**Strengths And Weaknesses:**

**Strengths**
- **Significance**: The paper studies an interesting problem that is very relevant to the ICLR community.
- **Presentation**: The presentation is clear and mostly easy to follow.
- **Soundness**:
    - The theoretical approach looks sound (I verified the arguments in the main paper, but did not assess the appendix in detail), and the result on the convergence of the representations is interesting, though not unexpected.
    - The experiments are designed well, and they support the main theoretical results (e.g. layer-wise convergence and prediction of answer preference changes).

**Weaknesses**
- **Originality (of the problem statement)**: I find the paper’s core results somewhat unsurprising: it is expected that representations stabilize after many repetitions of an answer, and that the residual streams are somewhat predictive of the model’s answer.
 - **Soundness of high-level conclusions**: A core high-level finding of the paper is that repeated contextual assertions should not be thought of accumulating evidence during inference, because some contextual assertions fail to alter a model’s prediction even in the limit of infinite repetition. I find this reasoning and the paper’s experimental demonstration of this incomplete (see in more detail below under ‘Connections to the Bayesian interpretation’).
- **Impact**: The answer shift-metric estimates how aligned are the per-layer residuals with the encouraged shift in the model’s answer. Its agreement with actual responses confirms the relevance of the residual streams, and validates the convergence analysis in the paper. However, beyond the mechanistic understanding, the practical benefits of the answer-shift metric are unclear, since computing the estimator requires computing intermediate representations of all layers, in which case one might as well compute the final prediction.

**Connections to the Bayesian interpretation**

The paper claims that repeated contextual assertions should not be thought of accumulating evidence during inference, because some contextual assertions fail to alter a model’s prediction even in the limit of infinite repetition. I find the paper’s reasoning and demonstration of this incomplete. Assuming a Bayesian view of in-context learning (Xie et al, 2022), the model’s initial likelihood of the wrong answer (assuming it is nonzero) determines how many repetitions are necessary to steer the model’s prediction towards the wrong answer. To my understanding, the paper’s theoretical results do not directly imply the existence of prompts/models for which no answer change happens at the infinite limit of repetitions. It could be that repetition induces slow drifts that gradually approach an answer change before settling to a limiting representation and answer.

In the paper’s experiments, answer changes are only reported in Figure 1 and Figure 4, where no answer change was only detected in two cases, with repetition counts of 100 and 1000, respectively. Initial likelihoods of the wrong answer were not reported / compared, hence the results do not rule out the case of an answer change for larger repetition count.

I recommend the authors address the connections of their results to Bayesian posterior convergence results, and cite the Bayesian ICL literature (Xie et al, 2022).

**Recommendation**

As per the current state of the paper, I recommend rejection, but I am open to reconsider if the authors address my concerns around the Bayesian interpretation.

**References**

Sang Michael Xie, Aditi Raghunathan, Percy Liang, Tengyu Ma. An Explanation of In-context Learning as Implicit Bayesian Inference. ICLR 2022. https://arxiv.org/abs/2111.02080

---

> ### Author Rebuttal · Authors · 2026-03-31
>
> We thank the reviewer for the helpful comments and respond below.
>
> ---
>
> > **Bayesian Interpretation**
>
> **1) Connection to Xie, et. al**
>
> Our setting is fundamentally different from Xie et al. (2022). They study increasingly many *independent* examples, each contributing new signal, whereas we study single-round repetition of the same assertion, with *no* new examples or evidence. Thus, their Bayesian account does not directly imply inevitable answer flips under infinite repetition.
>
> Meanwhile, our results do not contradict Xie et al.; rather, they suggest a different Bayesian abstraction for repetition-only prompts. Instead of a single shared concept for the full prompt, a more natural extension of Xie-style view is to use separate query and repetition concepts. Under this view, repetition can affect the prediction only within a bounded, query-dependent range and fully consistent with our results.
>
> We will add this discussion in our paper.
>
> **2) Non-flip cases N → $\infty$**
>
> We validate the existence of non-flip cases at N→ $\infty$ of Qwen-4B through the following two experiments across two settings (multi-choice & Yes/No).
>
> **Exp1: Limiting wrong-answer probability $P_{wrong, inf}$ in non-flip cases**
>
> We estimate $P_{wrong, inf}$ at N → $\infty$ via $P_{wrong, inf}=P_{wrong,init} + ΔP_{wrong} \le τ$, where $ΔP_{wrong}$ is
> the asymptotic shift in wrong-answer probability induced by repetition, as estimated from eq. (3). As shown, $P_{wrong, inf}$ remains below τ, supporting the existence of non-flip cases as N → $\infty$.
>
> **Table 1: Mean±std of $P_{wrong, inf}$, $P_{wrong,init}$, and $ ΔP_{wrong}$​ over non-flip cases**
> |Setting|$P_{wrong, init}$|$\Delta P_{wrong}$|$P_{wrong, inf}$|
> |---|---:|---:|---:|
> |Multi Choice, τ=0.25|2.67±2.02e-3|0.117±0.049|0.120±0.047|
> |Yes/No, τ=0.5|2.74±1.35e-3|0.139±0.078|0.141±0.078|
>
> **Exp2: Larger N validation of no-flip behavior**
>
> In **Table 2**, we report the $P_{correct}$ trajectories for four non-flip cases across two settings. Across all cases, $P_{correct}$ remains above the decision boundary and exhibits clear stabilization at large N.
>
> **Table 2: $P_{correct}$ trajectories for four non-flip cases**
> |**Question**|**0**|**500**|**1000**|**1500**|**2000**|**2500**|**3000**|**3500**|**4000**|**4500**|**5000**|**5500**|**6000**|**6500**|**7000**|**7500**|**8000**|
> |---|---|---|---|---|---|---|---|---|---|---|---|---|---|---|---|---|---|
> |9-156|0.99|0.59|0.78|0.87|0.62|0.62|0.84|0.87|0.95|0.88|0.88|0.84|0.88|0.87|0.85|0.87|0.84|
> |13-28|1.00|0.84|0.84|0.85|0.90|0.82|0.82|0.85|0.90|0.90|0.88|0.82|0.82|0.78|0.80|0.80|0.82|
> |10-1010|1.00|0.85|0.68|0.65|0.78|0.56|0.41|0.73|0.91|0.87|0.75|0.84|0.80|0.85|0.82|0.78|0.78|
> |10-1162|1.00|0.71|0.56|0.56|0.75|0.78|0.56|0.75|0.84|0.71|0.80|0.85|0.84|0.85|0.82|0.75|0.84|
> |**Mean**|**1.00**|**0.75**|**0.71**|**0.73**|**0.76**|**0.69**|**0.66**|**0.80**|**0.90**|**0.84**|**0.83**|**0.84**|**0.83**|**0.84**|**0.82**|**0.80**|**0.82**|
>
> **3) Exp3: Stability assessment at N=1000 vs. 2000**
>
> We also test N = 2000 in **Table 3** to address the concern that N=1000 may not be sufficiently large. The results remain close to those at N=1000, suggesting a stable regime by N=1000.
>
> **Table 3: Stability at larger N**
> |Model|Transfer Acc|Transfer F1|Mislead Acc|Mislead F1|Correct Acc|Correct F1|
> |---|---:|---:|---:|---:|---:|---:|
> |Falcon3-3B|94.0|96.8|93.8|96.7|91.1|95.2|
> |Qwen3-4B|97.1|98.5|90.6|94.6|93.4|96.5|
> |Mistral-7B|73.9|85.0|64.4|78.2|48.7|64.6|
> |**Average**|**88.3**|**93.4**|**82.9**|**89.8**|**77.7**|**85.4**|
>
> ---
>
> > **Q1: Flip step vs. $P_{wrong,init}$ and model size**
>
> We analyze when flips occur by grouping flipped cases into equal-count bins of $P_{wrong,init}$​ and reporting the mean flip step in each bin across three model sizes. **Table 4** shows that larger $P_{wrong,init}$ generally leads to earlier flips (Spearman $p<0.05$), while the dependence on model size is only partial.
>
> **Table 4: Flip step vs. $P_{wrong,init}$ (equal-sized bins)**
> |Model|$P_{wrong,init}$|Mean flip step|
> |---|---|---:|
> |Falcon3-3B|[0.01,0.06]|2.28|
> ||[0.06,0.09]|2.21|
> ||[0.09,0.13]|2.99|
> ||[0.13,0.19]|1.63|
> ||[0.19,0.46]|1.25|
> |Qwen3-4B|[0,0.003]|1.93|
> ||[0.003,0.01]|1.50|
> ||[0.01,0.03]|1.19|
> ||[0.03,0.08]|1.14|
> ||[0.08,0.47]|1.05|
> |Mistral-7B|[0.01,0.10]|5.18|
> ||[0.10,0.15]|1.77|
> ||[0.15,0.19]|1.21|
> ||[0.19,0.24]|1.07|
> ||[0.24,0.44]|1.60|
>
> ---
>
> > **Q2: Generality beyond RoPE**
>
> **Table 5** evaluates two non-RoPE models, BLOOM-7B (ALiBi) and GPT2-XL (absolute positional embeddings). Both show layer-wise stabilization at similar repetition counts, suggesting this is not RoPE-specific.
>
> **Table 5: Stabilization in non-RoPE architectures**
> |Model|α=0.1(A)|α=0.1(F)|α=0.05(A)|α=0.05(F)|
> |---|---:|---:|---:|---:|
> |BLOOM-7B|93.4|96.0|82.9|88.6|
> |GPT2-XL|97.2|95.7|93.8|91.1|
>
> ---
>
> > **Smaller Comments**
>
> We thank the reviewer for the helpful feedback and will improve the figure descriptions as well as the discussion of Xie et al. accordingly.

---

> > ### Author Rebuttal · Reviewer_JMWh · 2026-04-02
> >
> > Thank you for the rebuttal!
> >
> > Thank you for sharing your intuition about implicit Bayesian inference. I think that independence does not disqualify the Bayesian interpretation, since Bayesian inference can accommodate repeated observations. In your setting, the repeated contextual signal (“London is true”) consistently favors a particular answer, and under the implicit Bayesian inference view (e.g., Xie et al.), this repeated evidence should accumulate and eventually dominate the posterior, leading to a prediction change.
> >
> > However, the additional experiments support your claim that answer flips may never occur in some cases. I'm inclined to believe that there is something non-Bayesian going on, which is an interesting finding! However, this makes it all the more important to clarify what your theoretical results imply in the Bayesian context. There are many papers that dispute the empirical validity of the Bayesian interpretation, e.g. https://arxiv.org/abs/2406.00793. I wonder what your theoretical analysis adds to this debate?

---

> > > ### Author Response · Authors · 2026-04-05
> > >
> > > We thank the reviewer for acknowledging our additional experiments and for the follow-up, which highlights the **key implication of our theory**: _repeated assertions do not always accumulate into an eventual answer flip_.
> > >
> > > We agree that independence does not rule out a Bayesian interpretation. Our point is narrower: **existing Bayesian accounts of ICL rely on specific assumptions that do not automatically transfer to our setting**.
> > >
> > > ---
> > >
> > > >**Bayesian ICL Assumptions Matter Empirically**
> > >
> > > The Bayesian conclusion in Xie et al. is derived under a controlled relationship between the prompt distribution and known pretraining distribution (a mixture of HMMs). Their prompt is tied to the pretraining distribution via the assumption that the prompt concept belongs to the same family as pretraining (Asm. 4) and regularity conditions that keep the prompt compatible with the synthetic generative world (Asm. 5). This strict relationship is realized via their synthetic pretraining, where the model is trained on data generated from the same underlying distribution (Fig. 9).
> > >
> > > We show empirically that such assumptions materially affect behavior. Following the prompt–model coupling setting, we adapt the data construction and finetune the model so that the appended distractor becomes a reliable cue for its associated option. Under both True-to-False (T2F) and False-to-True (F2T), **Table 6 and 7** show significant changes in predictive distributions and correct/distractor probabilities.
> > >
> > > **Table 6 (T2F): Normalized $p_{correct}, p_{distractor}$, and distribution divergence — Base (B) vs. Finetune (F)**
> > > |Model|$p_c$(B)|$p_c$(F)|Sig|$p_d$(B)|$p_d$(F)|Sig|KL|Sig|JSD|Sig|
> > > |---|---:|---:|:---:|---:|---:|:---:|---:|:---:|---:|:---:|
> > > |Falcon3-3B|0.14|0.12|***|0.71|0.63|**|3.30|***|0.88|***|
> > > |Qwen3-4B|0.06|0.00|***|0.94|1.00|***|13.6|***|0.99|***|
> > > |Mistral-7B|0.04|0.01|***|0.89|0.97|***|0.73|***|0.76|***|
> > >
> > > **Table 7 (F2T): Normalized $p_{correct}, p_{distractor}$, and distribution divergence — Base (B) vs. Finetune (F)**
> > > |Model|$p_c$(B)|$p_c$(F)|Sig|$p_d$(B)|$p_d$(F)|Sig|KL|Sig|JSD|Sig|
> > > |---|---:|---:|:---:|---:|---:|:---:|---:|:---:|---:|:---:|
> > > |Falcon3-3B|0.16|0.09|***|0.69|0.56|***|3.38|***|0.88|***|
> > > |Qwen3-4B|0.01|0.00|***|0.99|1.00|***|14.4|***|0.99|***|
> > > |Mistral-7B|0.08|0.01|***|0.85|0.98|***|0.57|***|0.77|***|
> > >
> > > ---
> > >
> > > >**Problem Setting Matters Empirically**
> > >
> > > We next show problem setting matters. Falck et al. similarly support this claim that whether ICL appears “Bayesian” depends on the underlying settings, in their case i.i.d./exchangeable observation. We compare fixed- with diverse distractor repetition. Although both settings converge (**Table 8**), they converge to different predictive behavior (**Table 9**), with significant differences in output distributions and correct probabilities.
> > >
> > > **Table 8: Comparable convergence under fixed vs. diverse distractor repetition**
> > > |Model|α=0.1 (A)|α=0.1 (F)|α=0.05 (A)|α=0.05 (F)|
> > > |---|---:|---:|---:|---:|
> > > |Falcon3-3B|98.4|97.7|95.6|95.1|
> > > |Qwen3-4B|96.1|96.8|91.6|92.4|
> > > |Mistral-7B-v0.1|96.3|95.8|93.5|93.3|
> > >
> > > **Table 9: Distribution divergence and correct-answer probability under fixed vs. diverse distractor repetition**
> > > |Model|KL|Sig|JSD|Sig|$p_c$(Fixed)|$p_c$(Diverse)|Sig|
> > > |---|---:|:---:|---:|:---:|---:|---:|:---:|
> > > |Falcon3-3B|0.30|***|0.03|***|0.13|0.20|***|
> > > |Qwen3-4B|1.63|***|0.30|***|0.04|0.34|***|
> > > |Mistral-7B|0.97|***|0.12|***|0.04|0.09|***|
> > >
> > > ---
> > >
> > > >**Connecting Our Theory to the Bayesian View**
> > >
> > > **Our Novelty.** Our theory is derived from Transformer internal dynamics rather than Bayesian viewpoint. Our contribution to the Bayesian discussion is to provide a mechanism-level constraint: a separate theoretical lens, internal inference dynamics, for analyzing contextual sensitivity. This lens moves the analysis from output changes to internal predictive dynamics and links their convergence to asymptotic answer behavior.
> > >
> > > At a high level, our study aligns to a Bayesian latent-concept view of prediction. Conceptually, we study how repetition reshapes $p(θ∣S_n​,x_{test})$ and concept weighting accumulation mechanism. This perspective is consistent with the decomposition in Xie et al., whose Eqs. 5-6 can be written as
> > >
> > > $$p(y∣S_{n}​,x_{test}​)=∫​Σ_hp(y∣x_{test}​,h​,θ)p(h∣S_{n}​,x_{test}​,θ)exp(nr_n​(θ))p(θ)dθ$$
> > >
> > > and the concept weighting $exp(nr_n(θ))p(θ)$, which is proportional to the posterior $p(θ|S_n, x_{test})$ in
> > >
> > > $$p(y∣S_n​,x_{test}​)=∫​p(y∣S_n,x_{test}​,θ)p(θ∣S_n​,x_{test}​)dθ$$
> > >
> > > drives posterior accumulation under independent examples and leads to final flip.
> > >
> > > However, our convergence result in $p(y∣S_n,x_{test}​)$ with the bound of within-concept dependence of $p(y∣S_n,x_{test},θ​)$ on $S_n$ (Lem.1, Asm. 3), implies the saturation of $p(θ∣S_n​,x_{test})$ and further indicating the accumulation mechanism of concept weighting does not apply in our repetition-only setting. This provides a mechanistic explanation for the non-flip behavior, grounded in architecture-level convergence.

---

### Official Review · Reviewer_69jg · 2026-03-09

**Soundness:** 3
**Presentation:** 4
**Significance:** 3
**Originality:** 3
**Overall Recommendation:** 4
**Confidence:** 3

**Summary:**

This paper studies contextual repetition as a primitive behind many prompting/steering practices and asks: what happens asymptotically as repetition grows without bound? The major claim is that repeated contextual assertions do not behave as accumulating evidence during inference. Instead, internal predictive representations converge to stable, query-dependent regimes, implying that some answers are inherently resistant to contentual persuasion, while other queries exhibit inevitable flips. The authors provide a convergence analysis for decoder-only pre-norm Transformers with RoPE: RoPE attention logits are shown to have a bounded finite-frequency structure, enabling Cesàro-limit arguments for the repeated loop region and layer-wise convergence of representations at a fixed target suffix position as the number of repetitions grows. They then define a latent “answer-shift” metric using the output head difference vector and the residual stream at the prediction position, and propose a practical estimator of the infinite-repetition limit using Monte Carlo sampling over repeated loop blocks.

**Compliance With Llm Reviewing Policy:**

Affirmed.

**Final Justification:**

The rebuttal addressed my questions about this paper, reinforcing my prior assessment.

**Key Questions For Authors:**

- How sensitive are the convergence conclusions to Assumption 3.2 (uniform boundedness)? Can you empirically verify boundedness (or a weaker proxy) over the large N used in experiments?

- Do you expect similar convergence behaviour for non-RoPE positional encodings? If yes, what would replace the finite-frequency/Cesàro argument used here?

- What happens when the repeated block is paraphrased, reordered, or slightly perturbed each time (breaking strict periodicity)? Does the bounded drift claim still plausibly hold?

- You claim stable query-dependent limiting regimes constrain whether repetition can flip predictions. Can you predict regime membership from measurable quantities at N=0 (e.g., logit margin between candidate answers, entropy, calibration)?

**Limitations:**

The analysis is primarily about single-round, unbounded verbatim repetition in a fixed prompt structure. This is a useful primitive, but does not cover many forms of contextual influence in the wild (multi-turn dialogue, new evidence, paraphrase variation). Additionally,  the strongest theoretical claims rely on RoPE-induced oscillatory logits and periodicity. Extensions to other architectures/position schemes are not established

**Strengths And Weaknesses:**

Strength:

- Clear formalization of the unbounded repetition. The paper isolates repetition as the sole contextual signal (fixed query + repeated identical assertion block), removing confounds from extra evidence or multi-turn interaction.

- Motivation is strong and well-positioned. The paper frames a widely held prompting intuition (repetition accumulates evidence) and shows it is incomplete, with intuitive examples of heterogeneous behaviors (flip, delay, saturation, invariance).

- This paper is well-written, and I really enjoy reading this paper.

Weakness:

- The core regularity assumption is strong and under-discussed. Assumption 3.2 requires uniform boundedness of residual streams across all layers and repetition counts on relevant positions. This may be standard for analysis, but it is a big hinge for claims about real-world LLMs. The paper would benefit from a clearer discussion of when/why this is reasonable and whether empirical evidence supports boundedness under extreme repetition.

- Scope is tied to RoPE and the loop template repeated verbatim structure. The proofs lean on RoPE’s relative-position oscillatory structure and periodicity in the loop region. It is not yet clear how results extend to other positional schemes (ALiBi, learned absolute positions) or to soft repetition (paraphrase repetition) that breaks periodicity.

---

> ### Author Rebuttal · Authors · 2026-03-31
>
> We sincerely thank the reviewer for the thoughtful and constructive feedback, and address the comments below.
>
> > **Q1: Sensitivity and validation of boundness**
>
> Boundness is used in Theorems 3.5 and 3.6 to establish convergence across layers. Empirically, we compute the bound induced by the assumption 3.2 and verify it layerwise, finding the residual stream well bounded. We used a tighter bound defined as $C_0+\\sum_{j=0}^{L-1}(C_{A,j})$, where $C_0, C_{A,j}$ are the bound for the input and the attention residual state, in **Exp1** to validate the boundness across three models.
>
> **Exp1: Tightness and coverage of predicted bound**
> |Layers|Falcon3-3B Ratio|Falcon3-3B Cov|Qwen3-4B Ratio|Qwen3-4B Cov|Mistral-7B Ratio|Mistral-7B Cov|
> |---|---:|---:|---:|---:|---:|---:|
> |Front|0.27|1.00|0.46|1.00|0.28|1.00|
> |Mid|0.07|1.00|0.24|1.00|0.09|1.00|
> |Back|0.03|1.00|0.06|1.00|0.04|1.00|
> |**Mean**|**0.13**|**1.00**|**0.26**|**1.00**|**0.13**|**1.00**|
>
> ---
>
> > **Q2: Convergence of other position encodings**
>
> We further conducted experiments in models equipped with ALiBi (BLOOM-7B) and absolute positional encoding (GPT-XL) and observe the same convergence behavior as shown in **Exp2**.
>
> **Exp2: Convergence under ALiBi and absolute positional encodings**
> |Model|α=0.1 (A)|α=0.1 (F)|α=0.05 (A)|α=0.05 (F)|
> |---|---:|---:|---:|---:|
> |BLOOM-7B|93.4|96.0|82.9|88.6|
> |GPT2-XL|97.2|95.7|93.8|91.1|
>
> Theoretically, the finite-frequency/Cesàro argument is specifically needed for RoPE because RoPE introduces an oscillatory relative-position dependence into the attention logits. Other positional encodings lead to simpler situations and do not require the same machinery. Specifically, under absolute positional encodings, Eq. (2) reduces to the special case R(⋅)=1, while in ALiBi, the head output converges exponentially fast in N. So the convergence conclusion is unaffected without needing the finite-frequency/Cesàro argument.
>
> ---
>
> > **Q3: Convergence in paraphrased repetitions**
>
> We conducted experiments with paraphrased repetitions and found the convergence phenomenon maintained. Specifically, for multiple-choice questions, we randomly varied the option realization used in the repeated contextual assertion while preserving the same answer bias and measured layer-wise convergence under this paraphrased repetition setting in **Exp3**.
>
> **Exp3: Layer-wise convergence under paraphrased repetitions**
> |Model|α=0.1 (A)|α=0.1 (F)|α=0.05 (A)|α=0.05 (F)|
> |---|---:|---:|---:|---:|
> |Falcon3-3B|98.4|97.7|95.6|95.1|
> |Qwen3-4B|96.1|96.8|91.6|92.4|
> |Mistral-7B|96.3|95.8|93.5|93.3|
>
> This is consistent with prior work [1], which shows that iterative prompting with rephrased distractors leads to a bounded KL divergence. In our framework, convergence is a stronger stabilization property that implies such bounded behavior. Therefore, our finding of convergence under iterative rephrased distractors makes the phenomenon more coherent: prior work established the distribution-level consequence, while our result provides evidence for a stronger representation-level stabilization that helps complete the reasoning path.
>
> ---
>
> > **Q4: Predicting regime membership from measurable quantities**
>
> We agree that this is a very interesting and meaningful direction. At present, we cannot yet reliably predict regime membership directly from N=0 quantities alone such as the initial logit margin, entropy, or calibration.
> However, we do find encouraging evidence that the limiting regime becomes predictable once a modest amount of repetition is observed. Specifically, we use the behavior at N=500 to predict the regime at N=2000 across three different regimes and models, and obtain strong performance shown in **Exp4**.
>
> **Exp4: Predicting N=2000 regimes from N=500 behavior**
> |Model|Transfer Acc|Transfer F1|Mislead Acc|Mislead F1|Correct Acc|Correct F1|
> |---|---:|---:|---:|---:|---:|---:|
> |Falcon3-3B|66.9|80.0|82.4|90.2|88.5|93.8|
> |Qwen3-4B|84.3|91.5|75.9|85.3|82.7|90.5|
> |Mistral-7B|94.6|97.2|90.0|94.7|84.1|91.3|
> |**Average**|**81.9**|**89.6**|**82.8**|**90.1**|**85.1**|**91.8**|
>
> Lastly, we will stress our setting and discuss the limitations in our paper.
>
> ---
>
> > **Reference**
>
> [1] Yadkori, Yasin A., et al. "To believe or not to believe your llm: Iterative prompting for estimating epistemic uncertainty." Advances in Neural Information Processing Systems 37 (2024): 58077-58117.

---

> > ### Author Rebuttal · Reviewer_69jg · 2026-04-02
> >
> > Thanks for the additional experiments. My questions are resolved.

---

### Official Review · Reviewer_B75p · 2026-03-12

**Soundness:** 3
**Presentation:** 2
**Significance:** 3
**Originality:** 3
**Overall Recommendation:** 5
**Confidence:** 3

**Summary:**

This paper presents a theoretical framework for quantitatively analyzing how repetitions in context influence LM behavior and internal representation during inference. This framework predicts representation-level convergences under infinite repetitions, which can explain the persuasive power of that repetition (success or failure). They authors empirically test their insights on different LM families and find that they can accurately predict the LM's qualitative behavior under repetitive contexts.

**Compliance With Llm Reviewing Policy:**

Affirmed.

**Final Justification:**

Although this paper's results are on a very specific setting, I found the results to be insightful and interesting. I found the authors' answer to my questions/concerns to be satisfactory.

**Key Questions For Authors:**

1. Do you think your insights will hold if you were to randomly paraphrase some of the repetitions, instead of repeating the exact same context verbatim? I understand that the exact repetition setting allows for a clean theoretical analysis, but I am just curious about how much of these insights will hold under this minor change in the setting.

2. In Table 1 you show the fraction of layers whose ATTN/MLP representations have converged (under some tolerance) after 1000 repetitions. Did you try checking if these numbers varies across layer depth? For example, do you see more convergence in the early layers than in the later layers? or vice versa?

   * Another related question: you only present the results for N = 1000 repetitions. Did you check with N = 10, 100 etc.?

3. I would also be curious to know if you see different convergence patterns across different shifts (Transfer, Correct, Mislead) in Figure 3 and convergence values broken by layer depth. My hunch is that your Mislead (correct $\rightarrow$ incorrect) shift might be harder to converge, needing more repetitions and in later layers.

**Limitations:**

The authors should mention that this is a theoretical framework investigated in a very specific synthetic setting (repetitions of the same context).

**Strengths And Weaknesses:**

### Strengths
- I found the theoretical framework to be very insightful and well-motivated. Works on a real LM architecture.
- Although the problem set up can seem a bit contrived at first, I think this work addresses a real gap in our understanding that more repetition does not always lead to better persuation. And a mechanistic explanation of why this is the case.

### Weaknesses
- Works on a very specific setting (repetitions of the same context), which is a bit contrived.
- The paper is quite dense, which may make it difficult for some readers to follow. The presentation could be improved by providing more intuitive explanations and examples to illustrate the key concepts and results.

---

> ### Author Rebuttal · Authors · 2026-03-31
>
> We thank the reviewer for the constructive feedback and appreciate the positive comments.
>
> ---
>
> > **Q1: Convergence in paraphrased repetitions**
>
> We conducted experiments with paraphrased repetitions and found the convergence phenomenon maintained. Specifically, for multiple-choice questions, we randomly varied the option realization used in the repeated contextual assertion and measured layer-wise convergence under this paraphrased repetition setting in **Exp1**. The result is slightly lower than the repetition setting.
>
> **Exp1: Layer-wise convergence under paraphrased repetitions**
> |Model|α=0.1 (A)|α=0.1 (F)|α=0.05 (A)|α=0.05 (F)|
> |---|---:|---:|---:|---:|
> |Falcon3-3B|98.4|97.7|95.6|95.1|
> |Qwen3-4B|96.1|96.8|91.6|92.4|
> |Mistral-7B|96.3|95.8|93.5|93.3|
>
> This is consistent with prior work [1], which shows that iterative prompting with rephrased distractors leads to a bounded KL divergence. In our framework, convergence is a stronger stabilization property that implies such bounded behavior. Therefore, our finding of convergence under iterative rephrased distractors makes the phenomenon more coherent: prior work established the distribution-level consequence, while our result provides evidence for a stronger representation-level stabilization that helps complete the reasoning path.
>
> We thank the reviewer for raising this point, as it highlights a promising direction for future work.
>
> ---
>
> > **Q2.1: Convergence variance across different layers**
>
> We divided the layers into front / middle / back and measured convergence rates at sensitivity level α=0.05 and find a clear and consistent pattern across all three model families in  **Exp2**: earlier and middle layers show stronger convergence than later layers.
>
> **Exp 2: Layer-wise convergence by depth (α = 0.05)**
> |Model|A-front|A-mid|A-back|F-front|F-mid|F-back|
> |---|---:|---:|---:|---:|---:|---:|
> |Falcon3-3B|99.9|99.7|92.3|99.0|99.1|93.4|
> |Qwen3-4B|100.0|100.0|92.2|100.0|100.0|93.5|
> |Mistral-7B|100.0|100.0|97.1|100.0|100.0|96.1|
>
> Therefore, our experiments are consistent with the reviewer’s intuition: convergence is typically strongest in the earlier layers, whereas the later layers exhibit slightly greater variability.
>
>
> ---
>
> > **Q2.2: Convergence when N is limited**
>
> We evaluated convergence at N=10 and 100 and the rates are slightly lower than at N = 1000, suggesting that most layers for most questions are already close to their limiting regime at moderate repetition counts, while only a smaller subset of harder cases requires larger N to fully stabilize.
>
> **Exp 3: Layer-wise convergence of inference dynamics at N = 10**
> |Model|α=0.1 (A)|α=0.1 (F)|α=0.05 (A)|α=0.05 (F)|
> |---|---:|---:|---:|---:|
> |Falcon3-3B|97.9|99.4|94.5|97.3|
> |Qwen3-4B|99.5|99.4|96.3|96.9|
> |Mistral-7B|95.6|94.4|88.9|84.6|
>
> **Exp 4: Layer-wise convergence of inference dynamics at N = 100**
> |Model|α=0.1 (A)|α=0.1 (F)|α=0.05 (A)|α=0.05 (F)|
> |---|---:|---:|---:|---:|
> |Falcon3-3B|95.0|96.9|90.8|92.7|
> |Qwen3-4B|98.8|99.3|95.0|96.2|
> |Mistral-7B|88.5|86.6|80.8|74.9|
>
> ---
>
> > **Q3: Converge pattern across models and regimes**
>
> Our current observation is that the detailed convergence trajectory shows clearer differences across models than across regimes.
>
> For example, in Figure 3, the “rise-then-drop” behavior (e.g., the blue curve in Falcon) appears prominently in Falcon, but is not in the other models. By contrast, Mistral sometimes exhibits a sharper step-like increase followed by a rapid drop. At the same time, the majority of examples across models still follow the more common pattern of initial increase followed by near-stabilization.
>
> Thus, while different qualitative patterns do arise, we do not currently find a simple regime-level conclusion that generalizes across all models. Instead, the fine-grained convergence trajectory appears to be more strongly shaped by the model family itself. We agree this is an important direction for future study, and we will clarify this point in the revision.
>
> Lastly, per reviewer’s suggestion, we will stress our setting and discuss the limitations in our paper.
>
> ---
>
> > **Reference**
>
> [1] Yadkori, Yasin A., et al. "To believe or not to believe your llm: Iterative prompting for estimating epistemic uncertainty." Advances in Neural Information Processing Systems 37 (2024): 58077-58117.

---

> > ### Author Rebuttal · Reviewer_B75p · 2026-04-03
> >
> > I thank the authors for their response, which has adequately addressed my questions.

---

### Decision · Program_Chairs · 2026-04-30

**Decision:**

Accept (regular)

**Comment:**

This paper introduces a theoretical framework for analyzing how repeated contextual assertions influence the internal inference dynamics of decoder-only Transformers, proving that predictive representations converge to stable, query-dependent regimes rather than accumulating evidence without bound. Its main contribution is a principled, mechanism-level characterization of the limits of contextual persuasion, showing theoretically and empirically that some predictions are inherently resistant to repetition-based steering while others inevitably flip.

Strengths and weaknesses raised by reviewers:
- Strengths: Reviewers consistently found the theoretical framework well-motivated, insightful, and technically sound, addressing a real gap in understanding why more repetition does not always yield stronger persuasion (B75p, 69jg, ZBV4). The problem formulation cleanly isolates repetition as the sole contextual signal, and the empirical validation across multiple model families aligns well with the theoretical predictions (69jg, JMWh). The writing and presentation were generally viewed as clear and easy to follow, with a strong motivation grounded in common prompting intuitions (69jg, JMWh).
- Weaknesses: The setting is narrow — verbatim repetition of a fixed assertion — which reviewers noted is somewhat contrived and may limit generality to paraphrased, multi-turn, or non-RoPE scenarios (B75p, 69jg, JMWh). The core regularity assumptions (uniform boundedness, Lipschitzness, continuity in Assumption 3.2) are strong and initially under-discussed, with unclear empirical support (69jg, ZBV4). The connection to Bayesian interpretations of in-context learning (e.g., Xie et al.) is incomplete, and the strong abstract claim that repetition does not act as accumulating evidence is not fully supported by the theory as presented (JMWh). Finally, framing and writing issues remain, including grandiose language, biased related work coverage, and insufficient discussion of the interplay between pretraining/alignment and the observed convergence behavior (ZBV4).

Reviewer consensus after the rebuttal lands at 5/4/4/3, with the two strongest concerns (Bayesian framing and writing/framing) only partially resolved but the technical core and extensive new experiments (paraphrased repetitions, non-RoPE architectures, randomly-initialized models, boundedness verification, circuit-based analyses) satisfying the remaining reviewers; overall the paper offers a technically solid and novel mechanistic contribution worth sharing, despite needing revisions to soften central claims and improve framing.